# CONTROLLABLE CONTEXT SENSITIVITY AND THE KNOB BEHIND IT

**Julian Minder**[ð,ɔ,*]    **Kevin Du**[ð,*]    **Niklas Stoehr**[ð]    **Giovanni Monea**[ŋ]
**Chris Wendler**[ɔ]    **Robert West**[ɔ]    **Ryan Cotterell**[ð]
[ð]ETH Zürich    [ɔ]EPFL    [ŋ]Cornell University
jminder@ethz.ch  {kevin.du, niklas.stoehr, ryan.cotterell}@inf.ethz.ch
{chris.wendler, robert.west}@epfl.ch  giovanni@cs.cornell.edu

## ABSTRACT

When making predictions, a language model must trade off how much it relies on its context versus its prior knowledge. Choosing how sensitive the model is to its context is a fundamental functionality, as it enables the model to excel at tasks like retrieval-augmented generation and question-answering. In this paper, we search for a knob that controls this sensitivity, determining whether language models answer from the context or their prior knowledge. To guide this search, we first design a task for controllable context sensitivity. In this task, we feed the model a context, e.g., *Paris is in England*, and a question, e.g., *Where is Paris?*. Then, we instruct the model to either use its prior or contextual knowledge and evaluate whether it generates the correct answer for both intents, i.e., either *France* or *England*. When fine-tuned on this task, instruct versions of Llama-3.1, Mistral-v0.3, and Gemma-2 can solve it with high accuracy (85–95%). Analyzing these high-performing models, we narrow down which layers may be important to context sensitivity using a novel linear time algorithm. Then, in each model, we identify a 1-dimensional subspace in a single layer that encodes whether the model follows context or prior knowledge. Interestingly, while we identify this subspace in a fine-tuned model, we find that the exact same subspace serves as an effective knob in not only that model but also non-fine-tuned instruct and base models of that model family. Finally, we show a strong correlation between a model's performance and how distinctly it separates context-agreeing from context-ignoring answers in this subspace. Our results suggest that a single fundamental subspace facilitates how the model chooses between context and prior knowledge.

## 1 INTRODUCTION

Language models are often prompted with a query and preceding context, e.g., in retrieval-augmented generation or document analysis. In such applications, the language model needs to integrate information from both the context and its prior knowledge stored in its parameters. In some cases, we may prefer the model to rely more on the context, e.g., to avoid hallucinating responses based on outdated prior knowledge (Zhang et al., 2023); however, in other cases, we may prefer the model to rely more on its prior knowledge, e.g., to avoid being misled by misinformation provided in the context (Hong et al., 2024). As a motivating example, consider a document analysis application in which a language model is asked to help understand an opinion article in a newspaper. The user prompts the language model with the text of the article and then further prompts the model to summarize it with the query *What is the main argument of this article?*. In this case, the model should rely heavily on the context, i.e., the text of the article. Then, however, the user prompts the language model with the query *What are some criticisms of this argument?*. For the model to generate a useful response to this second prompt, the model cannot fully rely on the context of the article itself: an opinion piece may be written very authoritatively as if its conclusion was established fact, but still contains misleading claims in support of the writer's argument. Thus, in response to the second prompt, the language model should draw more upon its prior knowledge of the issue

---

[*]These authors contributed equally to this work.

and related opinions than blindly following the context. More broadly, because the optimal degree of context sensitivity depends highly on the use case, we contend that it is desirable to be able to specify how much and whether the model should be influenced by the context versus its prior knowledge.

Studies on the tension between context and prior knowledge have primarily focused on *knowledge conflicts* (Longpre et al., 2021), in which a given context directly contradicts information assumed to be in a model's prior knowledge about a query. For example, a language model trained on a sufficient amount of data should be able to reply to the query *What's the capital of France?* with *Paris*. However, if the context *The capital of France is London.* is prepended to the query, the model needs to decide whether to respond based on the context (*London*) or its prior knowledge (*Paris*). Prior studies (Longpre et al., 2021; Li et al., 2023b; Du et al., 2024; Monea et al., 2024; Ortu et al., 2024; Xie et al., 2024; Basmov et al., 2024) have shown that models will draw from context for some questions and prior knowledge from others. To investigate mechanisms underlying how the model draws from the context or prior knowledge, Yu et al. (2023), Ortu et al. (2024) and Jin et al. (2024) have searched for attention heads that promote each answer. However, these studies do not focus on whether or how the model deliberately mediates which source to rely on.

To this question of *how*, we hypothesize that there is a simple fundamental mechanism in the form of a subspace within the language model that facilitates the binary decision of whether to rely on the context or the prior knowledge. To guide our search for such a subspace, we design and execute a structured recipe. First, we create the **controllable context sensitivity (CCS)** task which augments the standard knowledge conflict setting with an *intent*, such as *Ignore the context* or *Listen to the context*. By disambiguating whether the model should follow context or prior knowledge through a simple addition to the prompt, we are able to identify and evaluate its behavior in both modes for the same context–query pair. We adapt models for this task using *fine-tuning* and *in-context learning*, then evaluate them on in-domain and out-of-domain test sets to assess whether they have developed a deeper ability to choose between context and prior knowledge beyond surface-level heuristics. In our case study on the Llama-3.1-8B family (Dubey et al., 2024), we find that both fine-tuning and in-context learning are moderately effective, with models excelling on in-domain test sets and significantly improving over zero shot baselines on out-of-domain test sets.

Armed with models that can perform the CCS task reasonably well, we then explore the mechanisms that facilitate their behavior in this task. Building on insights from Jin et al. (2024), we hypothesize that for a model to solve the CCS task, it must execute at least three high-level steps (in no particular order): extracting an answer from prior knowledge, extracting an answer from the context, and deciding to answer with the answer furnished by the context or the answer stored in its prior knowledge. We then seek to identify layers that may contain the model's computations that are aligned with each step. To do so, we develop an algorithm that uses tools from mechanistic interpretability to find a targeted subset of layers at which activation patching (Geiger et al., 2020; Vig et al., 2020; Meng et al., 2022) can switch a model from preferring the answer in the context to preferring the answer in its prior knowledge and vice versa. Then, building on ideas from distributed alignment search (Geiger et al., 2024), we identify a knob for the model's decision between following context or prior in the form of a 1-dimensional subspace. Despite locating such a knob on an instruct model fine-tuned on this task that states explicit intents, we show that it is even effective on non-fine-tuned and base models of the same family for prompts that do not state the intent.

Furthermore, we show strong evidence that for models good at the CCS task, the two intents correspond to two distinct values in that subspace, while bad models fail to exhibit this distinction. We repeat this process for Gemma-2 9B and Mistral-v0.3 7B to find a similar story. Our results suggest that a 1-dimensional subspace may be fundamental to many types of large language models (LLMs) in facilitating their ability to decide between following the context or their prior knowledge. These findings move toward developing more robust language models with controllable levels of reliance on context and prior knowledge. They further highlight how investigating models at a mechanistic level can yield high-quality interventions to control their behavior.

## 2 RELATED WORK

**Prior Knowledge in Language Models.** Prior studies have noted that LMs exhibit remarkable capabilities at answering questions depending on prior knowledge, such as factual recall. When queried, language models often generate plausible responses, indicating they may possess encoded

knowledge about entities (Brown et al., 2020; Petroni et al., 2019; Roberts et al., 2020; Geva et al., 2021). This knowledge is encoded in the model's weights as the model is exposed to mentions of these entities during pretraining (Xu et al., 2022; Zhou et al., 2023). Pretraining can lead to not only learning facts but also memorizing specific strings (Carlini et al., 2023; Stoehr et al., 2024b).

**Influence of Context on Language Models.** Models might also be prompted with context in addition to the query, which can be critical to the model solving the task effectively, such as in: (a) **In-context learning** (Brown et al., 2020), where demonstrations guide the model's response; (b) **Retrieval-augmented generation** (Lewis et al., 2020) and **open-book question-answering** (Mihaylov et al., 2018; Kasai et al., 2023), where relevant documents are included in context to aid query responses; (c) **Interactive dialogue/chat** (Vinyals & Le, 2015; OpenAI, 2023), where users converse with models over multiple turns; and (d) **Text annotation** (Ziems et al., 2024), where a model analyzes passages in the context for sentiment, toxicity, coherence, *inter alia*. However, other use cases may be better served by ignoring the context to some degree, i.e., in: (a) combating **jailbreaking** (Yu et al., 2024), e.g., ignoring attempts to override built-in model behaviors; (b) resilience to **misinformation** (Hong et al., 2024; Halawi et al., 2024), e.g., avoiding integrating incorrect information in the context; and (c) ignoring **irrelevant contexts** (Shi et al., 2023; Yoran et al., 2024). In all of these settings, models draw from two sources when responding: context, and knowledge encoded during training. Controlling context sensitivity in an application-dependent manner is key to robust use.

**Controlling Model Sensitivity to Context.** Several studies have proposed interventions to reduce dependency on prior knowledge and favor in-context information, including prompting (Zhou et al., 2023; Onoe et al., 2023), modifying training data (Wang et al., 2023a), fine-tuning (Li et al., 2023a), and activation-level interventions (Li et al., 2023c; Stoehr et al., 2024a; Yu et al., 2023; Ortu et al., 2024) at inference time. While Li et al. (2023a) aims for some level of controllable context sensitivity by attempting to ignore irrelevant context, they do not allow for explicit controllability. Neeman et al. (2023) train models to predict two answers using both context and prior knowledge. At a mechanistic level, Yu et al. (2023) and Ortu et al. (2024) use logit attribution methods (nostalgebraist, 2020) to inspect and identify attention heads which promote each answer. However, their interventions on these heads show limited bidirectional control, suggesting an inadequate capture of model behavior. Jin et al. (2024) uses path patching (Goldowsky-Dill et al., 2023; Wang et al., 2023b), an intervention-based method, to identify heads and show that zero-ablating a subset can effectively control model behavior.

**Identifying Mechanisms in Neural Networks.** According to the linear subspace hypothesis (Bolukbasi et al., 2016; Vargas & Cotterell, 2020; Wang et al., 2023c), model representations encode concepts as low-dimensional linear subspaces. Based on this hypothesis, prior work has explored how various concepts including truthfulness (Marks & Tegmark, 2024; Li et al., 2023c), humor (von Rütte et al., 2024), sentiment (Tigges et al., 2024), and refusal (Arditi et al., 2024) are encoded within model representations. Beyond identifying subspace representations, researchers have controlled model behavior by intervening on identified subspaces through additive steering, i.e., adding vectors to model representations (Rimsky et al., 2024; Turner et al., 2023; Zou et al., 2023; Ravfogel et al., 2022). Concept subspaces are commonly identified using distributed alignment search (Geiger et al., 2024), LEACE (Belrose et al., 2023b), mean and covariance matching (Singh et al., 2024), and difference in means (Marks & Tegmark, 2024).

## 3 HOW TO FIND THE KNOB BEHIND CONTEXT SENSITIVITY

### 3.1 DESIGNING THE TASK

First, we define the task of controllable context sensitivity based on *minimally contrastive* example pairs. Each pair has the same context $c$ and query $q$, differing only in whether the model should follow the context or prior knowledge. These pairs allow us to compare the model's internal states when it follows context versus prior knowledge, with all else equal.

Consider a language model $p$ over an alphabet $\Sigma$, i.e., $p$ is a distribution over the Kleene closure $\Sigma^*$. An element of $\Sigma$ is called a *token*. Further, consider a distinguished subset $\mathcal{Q} \subset \Sigma^*$ corresponding to licit queries and a distinguished subset $\mathcal{C} \subset \Sigma^*$ corresponding to licit contexts. Let $\varepsilon$ be the empty string. For a query $q \in \mathcal{Q}$, e.g., *What is the capital of France?*, and context $c \in \mathcal{C}$, e.g., *The capital*

*of France is London.*, let $a(\boldsymbol{q}, \varepsilon) \in \Sigma^*$ be the context-independent answer (*Paris*) and $a(\boldsymbol{q}, \boldsymbol{c}) \in \Sigma^*$ be the context-dependent answer (*London*). Let $w \in \{\text{ctx}, \text{pri}\}$ denote an **intent**, indicating whether to follow context (ctx) or prior knowledge (pri). Let $F: \mathcal{Q} \times \mathcal{C} \times \{\text{ctx}, \text{pri}\} \rightarrow \Sigma^*$ be a **formatting function** that maps a query, context, and intent to a formatted prompt, e.g., "*Context: The capital of France is London/n Instruction: Only listen to the context/n Query: What is the capital of France?*". Let $\mathcal{S}_{\text{trn}} \subset \mathcal{Q} \times \mathcal{C}$ and $\mathcal{S}_{\text{tst}} \subset \mathcal{Q} \times \mathcal{C}$ be disjoint training and testing sets of query–context pairs. Models are trained on $F(\boldsymbol{q}, \boldsymbol{c}, \text{pri}) \cdot a(\boldsymbol{q}, \varepsilon)$ and $F(\boldsymbol{q}, \boldsymbol{c}, \text{ctx}) \cdot a(\boldsymbol{q}, \boldsymbol{c})$ for $(\boldsymbol{q}, \boldsymbol{c}) \in \mathcal{S}_{\text{trn}}$, where $\cdot$ denotes concatenation.

## 3.2 IDENTIFYING MODEL BEHAVIOR

**Adapting a Model to this Task.** To study the model's mechanism, we first need it to controllably follow either context or prior knowledge. We adapt a language model to solve the task with two methods: (i) fine-tuning using a standard next-token prediction on the training set $\mathcal{D}_{\text{trn}}$, and (ii) using training samples as few-shot demonstrations for in-context learning.

**Evaluating Controllable Context Sensitivity.** We evaluate a model's ability to controllably choose between context and prior knowledge using *pair-accuracy*. An example is correct only if the model outputs the correct answer to a given query $\boldsymbol{q}$ and context $\boldsymbol{c}$ for both intents (ctx and pri), i.e., given a language model $p$ and dataset $\mathcal{S}$, with $\text{greedy}_{\boldsymbol{a} \in \Sigma^*}$ denoting the greedy decoding,

$$\text{PairAcc}(p, \mathcal{S}) \tag{1}$$
$$= \frac{1}{|\mathcal{S}|} \sum_{(\boldsymbol{q}, \boldsymbol{c}) \in \mathcal{S}} \mathbb{1}\{\text{greedy}_{\boldsymbol{a} \in \Sigma^*} p(\boldsymbol{a} \mid F(\boldsymbol{q}, \boldsymbol{c}, \text{ctx})) = a(\boldsymbol{q}, \boldsymbol{c})\} \mathbb{1}\{\text{greedy}_{\boldsymbol{a} \in \Sigma^*} p(\boldsymbol{a} \mid F(\boldsymbol{q}, \boldsymbol{c}, \text{pri})) = a(\boldsymbol{q}, \varepsilon)\}.$$

## 3.3 IDENTIFYING IMPORTANT LAYERS

Next, we need to identify layers in the model where the target behavior emerges. We focus on decoder-only transformer models (Vaswani et al., 2017). Building on prior work (Jin et al., 2024), we posit that for a model to succeed at this task, it must be able to execute at least three steps (not necessarily in this order): (i) extract the answer from the model's prior knowledge; (ii) extract the answer from the context; and (iii) decide whether to answer according to the context or the prior knowledge. Note that, under the framing of Geiger et al. (2024), these would be considered causal variables in a high-level model. Without specifying an exact causal graph, we argue these must be components in any reasonable one. We use tools from mechanistic interpretability to identify the layers at which the model appears to implement these steps.

**Intervention-based Interpretability.** Intervention-based interpretability techniques like activation patching are often used to identify which model activations are crucial for a task (Geiger et al., 2020; Vig et al., 2020; Meng et al., 2022). Intuitively, if intervening at some set of intermediate states can change a model's output behavior for a task, those intermediate states likely play a critical role in the model's ability in that task. Often, such interventions involve replacing intermediate states in the forward passes between two strings which differ minimally. For example, to identify activations that encode the *intent* of a prompt, we use two input strings that share the same *query* and *context* but differ in their *intent*. For a given model, $p$, we define a source string, $\boldsymbol{s} \in \Sigma^*$, and a target string, $\boldsymbol{t} \in \Sigma^*$. During the forward pass of $p(\cdot \mid \boldsymbol{t})$, we replace a subset of intermediate activations with those from $p(\cdot \mid \boldsymbol{s})$ and observe the effect on model internals and the output distribution of the patched $p(\cdot \mid \boldsymbol{t})$. We patch only at the last token, as prior work has shown this to be most informative for predicting the next token (Yu et al., 2023; Jin et al., 2024; Stoehr et al., 2024a; Monea et al., 2024). We also only patch the outputs of the multi-head attention (MHA) components in a transformer block; the intuition behind this choice is that this component ought to integrate information from the context into the *residual stream*—the hidden representation that each layer additively computes (Elhage et al., 2021)—of the last token. Interchanging these output activations allows us to analyze what kind of information is written on the residual stream and whether it has a causal effect on the model internals and the output distribution. By searching over different subsets of intermediate activations, we can identify those with the greatest impact on task performance.

Table 1: **Patching Setup**: To investigate the model's internal mechanisms, we use three distinct patching setups ($\mathcal{D}_w$, $\mathcal{D}_p$, and $\mathcal{D}_c$) to address our research questions. For all datasets, an example consists of a source prompt $s$, source answer $a_s$, target prompt $t$, and target answer $a_t$. $\mathcal{D}_w$ has two subvariants: $\mathcal{D}_w^{c \to p}$ and $\mathcal{D}_w^{p \to c}$, which represent different directions of the intervention.

| Dataset | Question & Description | $s$ | $a_s$ | $t$ | $a_t$ |
|---|---|---|---|---|---|
| $\mathcal{D}_w^{p \to c}$ | **Where is $w$ computed?** Only $w$ differs Source $w = $ pri | *The capital of France is London* **Ignore** *the context* *What is the capital of France?* | *Paris* | *The capital of France is London* **Only listen to** *the context* *What is the capital of France?* | *London* |
| $\mathcal{D}_w^{c \to p}$ | Only $w$ differs Source $w = $ ctx | *The capital of France is London* **Only listen to** *the context* *What is the capital of France?* | *London* | *The capital of France is London* **Ignore** *the context* *What is the capital of France?* | *Paris* |
| $\mathcal{D}_p$ | **Where is $a(q, \varepsilon)$ computed?** $w = $ pri, different $a(q, \varepsilon)$ | *The capital of France is London* *Ignore the context* *What is the capital of* **France**? | *Paris* | *The capital of France is London* *Ignore the context* *What is the capital of* **Italy**? | *Rome* |
| $\mathcal{D}_c$ | **Where is $a(q, c)$ computed?** $w = $ ctx, different $a(q, c)$ | *The capital of France is* **London** *Only listen to the context* *What is the capital of France?* | *London* | *The capital of France is* **Rome** *Only listen to the context* *What is the capital of France?* | *Rome* |

**Iteratively Searching For Important Components.** Searching for a small subset of MHA components at the last token position to patch is nontrivial because it is over an exponentially large space (i.e., $2^L$, where $L$ is the number of layers in the model) (Li et al., 2021). Thus, we use an iterative search algorithm to build a subset of important components, requiring $O(L)$ forward passes. In this algorithm, we use the *Token Identity Patchscope* (TIP) to observe model behavior at intermediate states (Ghandeharioun et al., 2024).[1] Specifically, we use it to identify the model's likelihood on the context and prior answers at intermediate layers and choose a subset of layers to patch that push the model to prefer the desired answer. Given a dataset of source and target pairs, the algorithm has two main steps. First, it identifies a continuous *base range* of layers where patching MHA components enables decoding the source answer from the residual stream at any layer. Then, it finds *inhibition layers* that suppress the source answer at later layers by iteratively patching MHA components until the source answer has a high probability at the last layer. We provide Python-esque pseudocode for our search algorithm in App. A.1.

**Patching Setups Per Subquestion.** We wish to address the three subquestions: (i) Where is the intent $w$ computed? (ii) Where is $a(q, \varepsilon)$ computed? (iii) Where is $a(q, c)$ computed? Answering each subquestion will demand applying the search algorithm described above on a specific patching setup, i.e., dataset, per subquestion. Each patching setup consists of tuples containing a source string, its associated answer, a target string, and the target's answer. The relationship between the source and target depends on the subquestion we aim to investigate. Table 1 outlines the specific patching setups for each subquestion. First, $\mathcal{D}_w^{c \to p}$ and $\mathcal{D}_w^{p \to c}$ hold the *context* and *query* constant but vary the intent $w$, enabling us to investigate how the model processes different intents. We define $\mathcal{D}_w^{p \to c} = \{(F(q, c, \text{pri}), a(q, \varepsilon), F(q, c, \text{ctx}), a(q, c))\}_{(q, c) \in \mathcal{S}_{\text{tst}}}$ and $\mathcal{D}_w^{c \to p} = \{(F(q, c, \text{ctx}), a(q, c), F(q, c, \text{pri}), a(q, \varepsilon))\}_{(q, c) \in \mathcal{S}_{\text{tst}}}$. $\mathcal{D}_p$ includes tuples where both the source and the target share the intent $w = $ pri, but differ in the prior answer $a(q, \varepsilon)$ they suggest, $\mathcal{D}_p = \{(F(q, c, \text{pri}), a(q, \varepsilon), F(q', c, \text{pri}), a(q', \varepsilon))\}_{(q, c) \in \mathcal{S}_{\text{tst}}, q' \in \mathcal{Q} \setminus \{q\}}$. This allows us to evaluate how patching alters the model's response with respect to $a(q, \varepsilon)$ and discern how the model computes $a(q, \varepsilon)$. Similarly, in $\mathcal{D}_c$ we explore how the model computes $a(q, c)$, $\mathcal{D}_c = \{(F(q, c, \text{ctx}), a(q, c), F(q, c', \text{ctx}), a(q, c')) \mid (q, c) \in \mathcal{S}_{\text{tst}}, c' \in \mathcal{C} \setminus \{c\}\}$.

### 3.4 IDENTIFYING THE CONTEXT-CONTROLLABILITY SUBSPACE FEATURE

**Learning the Context-versus-Prior Subspace.** Once we identified a subset of model components that potentially contain the mechanism for deciding between answering from the context or prior knowledge, we can further investigate whether this functionality can be encoded in a low-dimensional subspace within these components. According to the linear subspace hypothesis (Bolukbasi et al.,

---

[1]TIP interprets the information in a model's residual stream at intermediate layers by using the model to map from the residual stream at a given layer and token index to a distribution over tokens that best represents the information stored in that intermediate state. This approach can also be viewed as a variant of the SelfIE method (Chen et al., 2024). TIP outperforms other alternatives for interpreting intermediate states (e.g., probing (Tenney et al., 2019), LogitLens (nostalgebraist, 2020), and TunedLens (Belrose et al., 2023a)).

2016; Vargas & Cotterell, 2020), there exists a linear subspace $\mathcal{F} \subset \mathbb{R}^D$ which encodes the information about a specific concept. In our case, the concept of interest is whether the model uses the context or its prior knowledge. Because the CCS task involves a simple binary concept, we hypothesize that a rank-1 subspace encodes this concept. Informally, this hypothesis implies that a model's representation can be decomposed into a sum of orthogonal components, i.e., directions in space, and one such direction specifically encodes whether to follow the context or prior knowledge.

We use the algorithm presented in §3.3 to compute a *base range* of layers that appear to integrate the *intent* information. Let $\ell$ be the last layer in the *base range*. Let $\boldsymbol{h}^\ell \in \mathbb{R}^D$ denote the residual stream after layer $\ell$, i.e., the output of the $\ell^{\text{th}}$ transformer block at the last token position. We learn a rank-1 orthogonal projection matrix $\boldsymbol{P} \in \mathbb{R}^{D \times D}$ to project $\boldsymbol{h}^\ell \in \mathbb{R}^D$ onto a 1-dimensional subspace $\mathcal{F}_w$ of $\mathbb{R}^D$, encoding the intent $w$. We parameterize $\boldsymbol{P} = \boldsymbol{u}\boldsymbol{u}^\top$, where $\boldsymbol{u} \in \mathbb{R}^D$ is the basis vector of the subspace with a norm of 1; see App. F for a more detailed explanation of the parameterization of $\boldsymbol{P}$. Given a tuple $(\boldsymbol{s}, \boldsymbol{a_s}, \boldsymbol{t}, \boldsymbol{a_t}) \in \mathcal{D}_w^{p \to c} \cup \mathcal{D}_w^{c \to p}$, we define $\boldsymbol{h}_s^\ell$ to be the residual stream at the last token position after layer $\ell$ of the forward pass $p(\cdot \mid \boldsymbol{s})$, and similarly, $\boldsymbol{h}_t^\ell$ for $p(\cdot \mid \boldsymbol{t})$. To learn $\boldsymbol{P}$, we freeze the parameters of $p$ and patch the forward pass of $p(\cdot \mid \boldsymbol{t})$ as follows:

$$\boldsymbol{h}_t^\ell = (\boldsymbol{I} - \boldsymbol{P})\boldsymbol{h}_t^\ell + \boldsymbol{P}\boldsymbol{h}_t^\ell \qquad \textit{(normal decomposition)} \qquad (2a)$$

$$\widetilde{\boldsymbol{h}}_t^\ell \triangleq (\boldsymbol{I} - \boldsymbol{P})\boldsymbol{h}_t^\ell + \boldsymbol{P}\boldsymbol{h}_s^\ell \qquad \textit{(patched decomposition)} \qquad (2b)$$

Eq. (2a) expresses that we can decompose $\boldsymbol{h}_t^\ell$ into (i) the sum of the component representing our concept of interest ($\boldsymbol{P}\boldsymbol{h}_t^\ell$) and (ii) its orthogonal complement, the component which represents other information ($(\boldsymbol{I} - \boldsymbol{P})\boldsymbol{h}_t^\ell$). Then, in Eq. (2b), $\widetilde{\boldsymbol{h}}_t^\ell$ is constructed by replacing the component in $\boldsymbol{h}_t^\ell$ representing our concept of interest ($\boldsymbol{P}\boldsymbol{h}_t^\ell$) with the component in $\boldsymbol{h}_s^\ell$ representing the concept ($\boldsymbol{P}\boldsymbol{h}_s^\ell$). Thus, if $\boldsymbol{P}$ projects onto a subspace that encodes the *intent* concept, then the representation $\widetilde{\boldsymbol{h}}_t^\ell$ encodes the *intent* from $\boldsymbol{h}_s^\ell$ and all other aspects from $\boldsymbol{h}_t^\ell$. We visually illustrate these decompositions in App. G.

We denote $\widetilde{p}_\ell(\cdot \mid \boldsymbol{t}; \boldsymbol{P}, \boldsymbol{s})$ to be the language model with activation $\boldsymbol{h}_t^\ell$ replaced by $\widetilde{\boldsymbol{h}}_t^\ell$ as defined in Eq. (2b). We construct a training set $\{(\boldsymbol{s}_n, \boldsymbol{a}_{\boldsymbol{s}_n}, \boldsymbol{t}_n, \boldsymbol{a}_{\boldsymbol{t}_n})\}_{n=1}^N \subset \mathcal{D}_w^{p \to c} \cup \mathcal{D}_w^{c \to p}$. As can be seen in Tab. 1, this dataset contains matched pairs $(\boldsymbol{s}_n, \boldsymbol{t}_n)$ which differ only in the specified intent. Then, to learn $\boldsymbol{P}$ which well-represents our concept, we minimize the following objective over the training set:

$$J_\ell(\boldsymbol{P}) = -\frac{1}{N} \sum_{n=1}^{N} \log \widetilde{p}_\ell(\boldsymbol{a}_{\boldsymbol{s}_n} \mid \boldsymbol{t}_n; \boldsymbol{P}, \boldsymbol{s}_n) \qquad (3)$$

That is, we minimize the cross-entropy loss between the language model when patched with $\widetilde{\boldsymbol{h}}_t^\ell$ and the label $\boldsymbol{a_s}$. Since $\boldsymbol{s}_n$ and $\boldsymbol{t}_n$ always have different intents $w$, but share the same *context* and *query*, we are effectively optimizing for a subspace where replacing the subspace component of $\boldsymbol{t}_n$ with the corresponding component of $\boldsymbol{s}_n$ leads to an answer that reflects the flipped intent.

**Controlling Model Behavior Using the Context-versus-Prior Subspace.** After learning an orthogonal projection matrix $\boldsymbol{P}$ to project a vector into the context-versus-prior subspace, we can control the model's behavior by setting the subspace component based on the input intent $w$. To do this we define a *function* $c : \{\text{ctx}, \text{pri}\} \to \mathbb{R}$ that acts as a scalar for the basis vector $\boldsymbol{u}$ of $\mathcal{F}_w$ and returns a constant corresponding to one of the two intents. The resulting *patched decomposition* is defined as:[2]

$$\widetilde{\boldsymbol{h}}_t^\ell \triangleq (\boldsymbol{I} - \boldsymbol{P})\boldsymbol{h}_t^\ell + \boldsymbol{P}\boldsymbol{u}c(w) \qquad (4)$$

The function $c$ represents the knob to steer which behavior to follow. A successful static intervention on a learned subspace $\mathcal{F}_w$ implies that we have not only identified a 1-dimensional subspace representing intent but also determined how to manipulate it manually. We evaluate the effectiveness of a static intervention using the *pair-accuracy*.

## 4 CASE STUDY: LLAMA-3.1 8B

We describe detailed results in executing the recipe from §3 to identify the mechanism behind controllable context sensitivity. Results for additional models are in §5 and App. H.

---

[2]$\boldsymbol{P}$ is redundant in the second term of Eq. (4) since $\boldsymbol{P}\boldsymbol{u}c(w) = \boldsymbol{u}\boldsymbol{u}^T \boldsymbol{u}c(w) = \boldsymbol{u}c(w)$, but is included for consistency.

## 4.1 TASK SETUP

**Datasets.** Following the task formulation in §3.1, we construct intent-augmented datasets, CCS-BF, CCS-MH, and CCS-AR, based on the query-context pairs in BASEFAKEPEDIA, MULTIHOPFAKE-PEDIA (Monea et al., 2024), and ARITHMETIC. BASEFAKEPEDIA is a knowledge conflict dataset from Wikipedia with queries across 23 relation types (e.g., *Norway's capital city* or *Mac Pro, a product created by*) and paragraphs generated by a language model that provide counterfactual answers. MULTIHOPFAKEPEDIA resembles BASEFAKEPEDIA but requires an extra hop of reasoning (e.g., *London is the capital of France. Tunis is in the same country as London. What country is Tunis in?*). ARITHMETIC is a synthetically generated dataset whose queries are simple arithmetic expressions using the operators $\{+, -, \times, \div, \exp\}$ and contexts are reassignments of subexpressions to another value resulting in a counterfactual answer. For example, given the query *(5 + 1) / 2 =* and the context *5 = 9*, the prior answer would be *3*, while the context answer would be *5*. We limit expressions to a depth of 2, i.e., two operators, with input and output numbers between 0 and 9.

**Intent Format.** We also format the intent $w \in \{\text{ctx}, \text{pri}\}$ in two different ways to probe the model's robustness to different formulations of the same intent. First, the *instruction* intent format (🔶) expresses the intent as a string instruction, e.g., *Ignore the context in answering the query.* or *Only consider the context in answering the query.* Second, the *weight* intent format (🔢) expresses the intent as a context weight, e.g., *Context weight: 0* or *Context weight: 1*.

## 4.2 ADAPTING MODELS TO THE TASK

**Training.** We adapt the instruct Llama-3.1 8B (Dubey et al., 2024) to this task in two ways: (i) QLoRA fine-tuning (FT) the attention components using CCS-BF's training set, and (ii) in-context learning (ICL) with 10 prepended CCS-BF examples. Training details are in App. D.

**Evaluation.** We examine two forms of generalization: robustness to different datasets, and robustness to different intent formats. For the former, we test whether a model trained on CCS-BF can perform well on test splits from CCS-BF, CCS-MH, and CCS-AR. For the latter, we assess whether a model trained with one intent format, e.g., 🔶, performs well with prompts in another format, e.g., 🔢.

**Results.** Fig. 1 shows the generalization results for Llama-3.1-8B-Instruct. The model achieves high pair accuracy on its in-domain test set with FT ($\approx 90\%$) and ICL ($\approx 88\%$). However, performance drops significantly for ICL and mildly for FT on CCS-MH, which requires additional reasoning. On CCS-AR, both models show significant degradation, as the task is out-of-domain and demands reasoning beyond context extraction. Fig. 1b shows that, for intent formats, the model: (i) performs well when fine-tuned on either intent format, (ii) generalizes well from the 🔢 to the 🔶 format, and (iii) struggles when trained on the 🔶 format but evaluated on 🔢. This result is intuitive as the instruct model is tuned to follow natural language instructions such as 🔶, but may not be familiar with interpreting the 🔢 instruction. Overall, the model (i) learns the task in-domain with high accuracy, (ii) generalizes moderately well to other datasets, depending on the degree of difference, and (iii) adapts reasonably well to other intent formats, especially if they are in natural language.

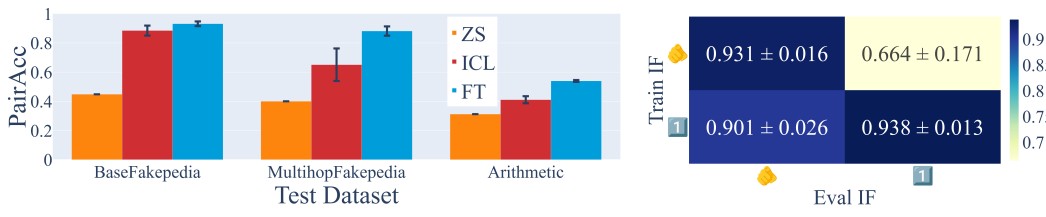

(a) Generalization to Datasets.

(b) Generalization to Intent Formats (IF).

Figure 1: (a) Pair accuracy of Llama-3.1-8B-Instruct when trained on CCS-BF and evaluated on CCS-BF, CCS-MH, and CCS-AR datasets. For each dataset, we evaluate the model zero-shot, with 10 in-context learning examples from CCS-BF, and after fine-tuning on 2048 examples from CCS-BF. (b) Pair accuracy when trained and evaluated on different intent formats, where 🔢 and 🔶 mean the intent is expressed as a numerical context weight or as a string instruction, respectively.

### 4.3 IDENTIFYING IMPORTANT COMPONENTS

Focusing on Llama-3.1-8B-Instruct fine-tuned using the intent format 🍯, we apply the algorithm presented in §3.3 to identify important layers that appear to facilitate the model's sensitivity to context. First, we investigate where the intent $w$ is computed by using tuples from $\mathcal{D}_w^{p \to c}$ and $\mathcal{D}_w^{c \to p}$, described in Tab. 1. Second, we investigate which layers compute the prior answer $a(q, \varepsilon)$ and context answer $a(q, c)$. If these layers are later than the ones identified in the first step then that could suggest that $w$ is encoded in the residual stream and depending on its value, either $a(q, c)$ or $a(q, \varepsilon)$ is retrieved.[3]

#### 4.3.1 WHERE IS $w$ COMPUTED?

We aim to identify where the model initially incorporates information about the intent and how this affects its predictions. We use the algorithm described in §3.3 and App. A.1 on both $\mathcal{D}_w^{c \to p}$ and $\mathcal{D}_w^{p \to c}$ and report the identified layers in Fig. 2a and Fig. 2b, respectively. We observe that in both directions, patching the MHA outputs for layers 12 to 16 suffices to switch the prediction from agreeing with the context (CTX) to agreeing with the prior (PRIOR) and vice versa. This suggests two hypotheses: either these layers load the correct answer into the residual stream, or they encode the intent $w$, which subsequently triggers the loading of the correct answer in later layers. However, Fig. 2b shows the model has a low probability of the context answer until after layer 24, supporting the latter hypothesis.

#### 4.3.2 WHERE ARE $a(q, \varepsilon)$ AND $a(q, c)$ COMPUTED?

We apply the same algorithm to $\mathcal{D}_p$ and $\mathcal{D}_c$ to identify which layers load the two answers, $a(q, \varepsilon)$ and $a(q, c)$. For $\mathcal{D}_p$, we patch activations from a source (SRC PRI) into a target (TGT PRI), both sharing the same intent pri but different prior answers $a(q, \varepsilon)$. For $\mathcal{D}_c$, we patch from a source (SRC CTX) into a target (TGT CTX), both having intent ctx but different context answers $a(q, c)$. Fig. 2c and 2d show the layers found for the prior answer and the context answer, respectively. In ablation studies Fig. 7a, we show that the context answer can be also integrated with only layers after layer 23. Since the prior answer (Fig. Fig. 2c) is integrated at different layers than the context answer ( 2d and 7a), distinct mechanisms likely handle each answer. Layer 24 seems crucial in both processes. Ablation studies in App. A.2 show that neither $a(q, \varepsilon)$ nor $a(q, c)$ can be effectively patched without layer 24 (Fig. 7b and 6c). We hypothesize layer 24's role varies by intent, conditionally loading either the prior or context answer. Since the model's preference for context or prior answer stabilizes after layer 16, this suggests that the intent is encoded after this point and later layers such as layer 24 read it. Given the binary nature of the intent variable, we hypothesize that its encoding can be modified to selectively trigger the loading of either the context or prior answer.

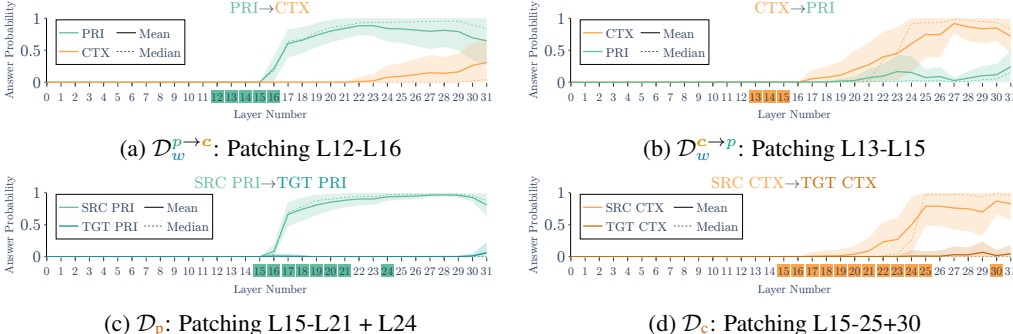

(a) $\mathcal{D}_w^{p \to c}$: Patching L12-L16

(b) $\mathcal{D}_w^{c \to p}$: Patching L13-L15

(c) $\mathcal{D}_p$: Patching L15-L21 + L24

(d) $\mathcal{D}_c$: Patching L15-25+30

Figure 2: Answer probabilities per layer as determined by TIP for different patching settings on Llama 3.1 Instruct FT 🍯. The $x$-axis represents the layers. The $y$-axis shows the TIP answer probability. On the $x$-axis we mark the patched layers. Each row of subplots aims to answer one subquestion. Top Row: **Where is $w$ computed?** Patching a source SRC PRI into a TGT CTX (left; 2a) and vice versa (right; 2b). Bottom Row: **Where is $a(q, \varepsilon)$ and $a(q, c)$ computed?** Patching a source SRC PRI into a TGT PRI, using samples from $\mathcal{D}_p$ (2c) and the same for CTX (2d).

---

[3]We have previously reported slightly different layers for Fig. 2b to 2d, which were reflecting true model behavior but were found manually, not by Listing 1. Previously reported plots are Fig. 7a, 7d and 7e.

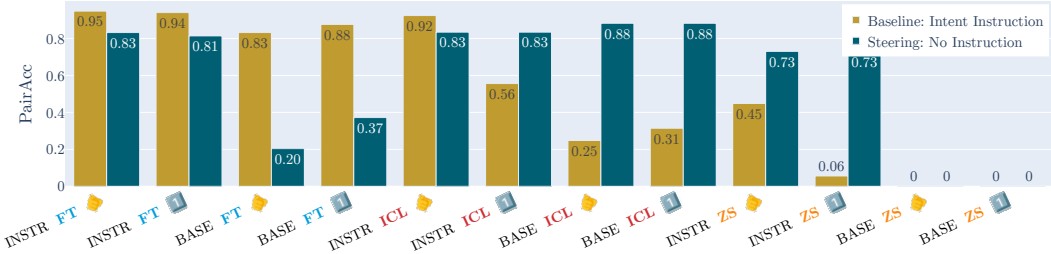

Figure 3: The baseline (yellow) reflects the *PairAcc* of a model evaluated on CCS-BF without steering. In blue we show the *PairAcc* when the explicit intent instruction is removed and the subspace $\mathcal{F}_w$ manually set. Although $\mathcal{F}_w$ was learned for INSTR FT 🪙, it transfers well to other configurations, as evidenced by the blue bar approaching or exceeding the yellow for most configurations.

### 4.4 Identifying the Context-Controllability Subspace

Following §3.4, we learn a rank-1 orthogonal projection matrix $P$ to identify a subspace $\mathcal{F}_w$ encoding intent. We search for this subspace in *layer 16*, as this is the last layer in the *base range* of influential layers found in §4.3.1 using the algorithm described in §3.3. We train on the subset of $\mathcal{D}_w^{p \to c} \cup \mathcal{D}_w^{c \to p}$ of CCS-BF for which the model answers correctly for both intents. If this subspace indeed controls the choice between context and prior, then we should be able to remove the intent from the input and still steer the model to produce the intended output by setting the value of $c(w)$ according to Equation 4. For these interventions, we choose $c(\text{pri}) = -6$ and $c(\text{ctx}) = 6$ based on performance on a validation set. For example, a model should be able to answer *The capital of France is London. What is the capital of France?* with *London* when steered with $c(w) = 6$ and *Paris* when $c(w) = -6$.

Fig. 3 shows that the identified subspace strongly aligns with the causal variable for intent, allowing for effective model steering. On the fine-tuned instruct model, we achieve $83\%$ *PairAcc* using steering, compared to the $95\%$ baseline (very left; INSTR FT 🪙). This is notable, given we manipulate only a 1-dimensional subspace in a single layer. Additionally, the figure shows that this same subspace aligns well with the causal variable for intent across different model configurations. We successfully transfer $\mathcal{F}_w$ to both the non-fine-tuned Llama-3.1-8B-Instruct (INSTR) and the base Llama-3.1-8B (BASE) model. The subspace performs particularly well on the base model in the in-context learning (ICL) setting, where *PairAcc* significantly exceeds the baseline accuracy as well as the steered fine-tuned model. Moreover, we highlight the zero-shot (ZS) performance of the instruct model ($73\%$), significantly outperforming the baseline performance. However, the ZS performance on the base model results in $0\%$ *PairAcc*, as the model lacks training for instruction-following tasks. While the subspace intervention is relatively ineffective on the fine-tuned base model, we hypothesize that this is because the weights of this model are likely the furthest from those of the fine-tuned instruct model.

## 5 A Fundamental Subspace for Controllable Context Sensitivity

Due to the strong evidence for a high alignment of $\mathcal{F}_w$ to the causal intent variable, we propose two hypotheses: (i) This subspace is fundamental to the model and different learning methods learn to set the value of this subspace. (ii) As a fundamental subspace to language models, a similar rank-1 subspace to encode choosing context or prior knowledge can be found in other language models, too.

We provide evidence to support hypothesis (i). First, Fig. 3 shows that adjusting the value of the subspace can recover or even surpass baseline performance in both fine-tuned and non-fine-tuned models. Notably, the exceptional efficacy of the subspace intervention in the zero-shot evaluation of the instruct model—which has never seen examples of this task—suggests that this capability is already present in the model and can be activated by setting $\mathcal{F}_w$. Second, Fig. 4a shows that the subspace generalizes to multiple out-of-domain datasets, with steering performance either competing with or surpassing the intent instruction baseline across different datasets. This holds for not only the fine-tuned instruct model but also ZS evaluations on the instruct model and ICL on the base model. Finally, we find a strong, statistically significant correlation (0.908) between a model's performance and how well it distinguishes values in that subspace with different intents in the prompt. As displayed in Fig. 5, the difference in subspace value when the intent is pri vs ctx tends to be higher for better models at this task. This suggests that well-performing models know to set this value in the subspace.

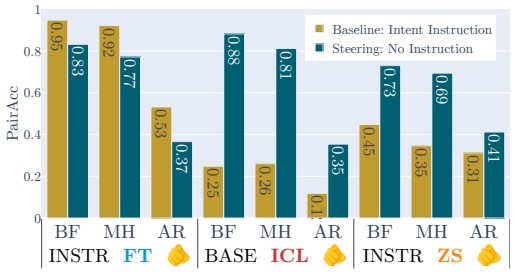
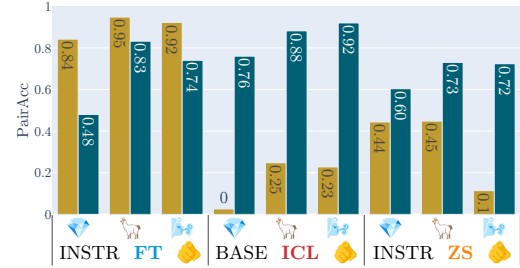

(a) Llama 3.1: Other Datasets
(b) BaseFakepedia: Other Models

Figure 4: We compare pair accuracy of a baseline model (with intent instructions) against the steered model (without intent instructions). We consider baseline models: (i) instruct model fine-tuned on CCS-BF, (ii) base model with 10 in-domain ICL demonstrations, and (iii) the default instruct model. Left: Subspace steering on Llama 3.1 generalizes across datasets (BASEFAKEPEDIA (BF), MULTIHOPFAKEPEDIA (MF), and ARITHMETIC (AR)). Right: For multiple models (Llama 3.1 8b (🦙), Gemma 2 9b (🔷), and Mistral 7b v0.3 (🌀)), a rank-1 subspace can be used for effective steering.

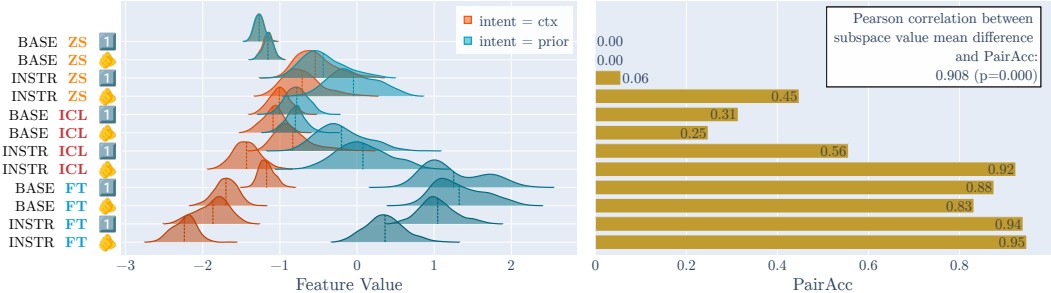

Figure 5: Subspace $\mathcal{F}_w$ value distributions of different model configurations (left) and baseline model performance on CCS-BF (right). We can observe a high correlation between the absolute difference between the means of the two groups (ctx and pri) and the performances.

We also identify the described subspace in Gemma-2 9B (Riviere et al., 2024) and Mistral-v0.3 7B (Jiang et al., 2023), using the same methodology. Fig. 4b shows that, for each model family, their respective subspaces are transferrable from the fine-tuned instruct model to both the non-fine-tuned instruct model and the base model. In App. H we provide a detailed study of the subspace in other models, including a high correlation between model performance and subspace values.

## 6 DISCUSSION, LIMITATIONS, AND FUTURE WORK

While our study presents evidence that a model can be induced to controllably draw from context or prior knowledge in answering questions in these specific settings, it is important to characterize the nature of the exact model capability we are examining in this study. In particular, both fine-tuning and turning this knob for a model seem to be more effective when the model can directly copy the answer from the context (when the intent is ctx). For example, in the ARITHMETIC task, a context might explicitly contain the answer, e.g., *(5 + 1) / 2 = 7*, or it might only override a subproblem, e.g., *5 + 1 = 8*. Generally, the models are better at producing the context-agreeing answer when it is explicitly stated in the context. More investigation is needed to understand to what extent a model can use information from context as part of an intermediate reasoning chain as opposed to direct copying.

Zooming out, our work highlights the importance of studying the fundamental functionality in language models of controllable context sensitivity. We show how tools from mechanistic interpretability can be useful toward both understanding how models implement this functionality and controlling the behavior; further, such an approach could help understand mechanisms behind other functionalities. Promising future directions include: (i) evaluating whether this subspace influences additional behaviors like instruction-following, (ii) learning to adaptively steer, i.e., the model automatically decides when it should leverage or ignore context (especially in settings such as retrieval-augmented generation), and (iii) beyond traditional knowledge conflicts, developing datasets that involve integrating information from both context and prior knowledge rather than only choosing between the two.

## CONTRIBUTIONS

Julian Minder designed and conducted the interpretability experiments presented in §4 as well as the subspace analysis in §5. The formalization of the interpretability-related methodology, in particular the subspace analysis, was developed in collaboration with Kevin Du. Kevin Du coordinated the project, implemented the code for training and evaluating language models discussed in §4.2, ran all training experiments and conducted the quantitative evaluations. Kevin Du, Niklas Stoehr and Chris Wendler started the project based on initial ideation and proposed the controllable context sensitivity task as a follow up to prior work of Kevin Du, Niklas Stoehr and Ryan Cotterell. Niklas Stoehr, Giovanni Monea, Chris Wendler, Robert West and Ryan Cotterell advised on the methodological design and contributed to the conceptualization of the work throughout. In particular, Giovanni Monea advised on data matters sharing his experience from creating the Fakepedia dataset and Chris Wendler helped jump-start the model fine-tuning by sharing code.

## ETHICS STATEMENT

As LLM capabilities grow more advanced and their usage proliferates throughout the real world, we acknowledge that their development can exacerbate risks to people via misinformation or hallucination, especially those historically underrepresented or misrepresented to these models. Our work aims to make model behavior more transparent by providing a new tool to analyze the interaction between context and prior knowledge in LMs, which is especially important as people interact with them in chat, question-answering, and other prompt-based settings. We foresee no particular ethical concerns and hope this paper contributes to developing tools that can identify and mitigate ethical concerns in the future.

## REPRODUCIBILITY STATEMENT

We provide code to reproduce all datasets, experiments, and analysis at `https://github.com/kdu4108/context-vs-prior-finetuning`.

## ACKNOWLEDGEMENTS

Niklas Stoehr acknowledges funding through the Swiss Data Science Center (SDSC) Fellowship. Robert West's lab is partly supported by grants from the Swiss National Science Foundation (200021_-185043,TMSGI2_211379), Swiss Data Science Center (P22_08), H2020 (952215), and by generous gifts from Meta, Google and Microsoft. We also thank Mike Chen for pointing out a typo in Listing 1.

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

## A    SEARCHING FOR IMPORTANT LAYERS

### A.1    ALGORITHM

We describe the algorithm in Python-esque pseudocode in Listing 1. For more details on the Token-Identity Patchscope (TIP) method (PATCHSCOPE), see Ghandeharioun et al. (2024). For more details on activation patching (INTERCHANGE), see Meng et al. (2022). In Fig. 6 we visualize the TIP at different stages in the algorithm.

The goal of this algorithm is to find a subset of layers for which patching the MHA output from the forward pass of a source example into that of a target example results in the desired effect, i.e., the source answer being decoded with a significantly higher probability than the target answer. On one extreme end, patching all layers replicates the source forward pass, ensuring the desired effect (assuming the patched last token is the same between source and target examples). Conversely, with no patching, the forward pass remains equivalent to the target forward pass.

In step 1, we aim to determine a base range of layers. When this range is patched, the source answer should appear with high probability at some intermediate layer—not necessarily the last one. Fig. 6c illustrates the base range patched for $\mathcal{D}_p$. Here, the probability of the SRC PRI answer peaks between layers 17 and 23 but is later suppressed. We identify this base range by first finding its upper bound, end_l (Step 1.1). We incrementally patch layers from 0 to end_l until the source answer achieves high probability at a specific layer, as shown in Fig. 6b. Next, we adjust the lower bound, start_l, until increasing it further causes a drop in the maximum probability of the source answer. This defines our base range.

If patching only this base range already elevates the source answer's probability significantly higher than the target answer's at the output, the process is complete. Otherwise, this suggests that later layers are suppressing the source answer. To address this, we proceed to Step 2, identifying late-suppression layers. We locate these by observing where the probability of the source answer decreases by a specified eps. We then patch these layers iteratively until the source answer's probability exceeds the target's by the required margin. As demonstrated in Fig. 6d, for $\mathcal{D}_p$, patching the MHA output of the late-suppression layer 24 alone suffices to achieve the desired effect.

### A.2    ABLATIONS IN SEARCHING FOR IMPORTANT LAYERS (LLAMA-3.1)

We run ablations to identify the importance of different layers in Llama-3.1. Fig. 7 shows additional experiments and alternative solutions, demonstrating that multiple sets of layers can achieve the same goal. From Fig. 7b, we can see that without patching layer 24 for SRC CTX, the alternate context answer never becomes the top-probability answer at any layer according to the TIP. This suggests layer 24 is critical for loading in the context answer, especially as it also acts as a late-suppression layer for the prior. In Fig. 7a, we show that the context answer can also be integrated with patching only post 24 layers. From Fig. 7c, we see that only patching in layers 15-16 in an attempt to make the model respond with a SRC PRI fails to significantly raise the probability of the SRC PRI at any layer. This suggests that layers after layer 16 are also critical to loading in the prior answer.

## B    MLP DISCUSSION

Recent studies have shown that prior knowledge in Transformer models is primarily stored in MLP weights (Meng et al., 2022; Geva et al., 2021; 2022; Dai et al., 2022). This raises the question of why MLPs are not central to our investigation. Mechanistic analyses from recent works (Jin et al., 2024; Geva et al., 2023) suggest that MLPs in earlier token positions extract answers, which are then relayed to the final position via attention heads. Thus, the MLPs at the last token position contribute minimally to direct answer computation. Ortu et al. (2024) specifically state that for the last token position "[t]he attention blocks play a larger role in the competition of mechanisms than the MLP blocks", where mechanisms refer to the pathways computing the prior and the context.

We tested our hypothesis by patching the MLP outputs across all layers using the $\mathcal{D}_p$ setup. We anticipated that if the MLPs at the final token position were crucial for determining the prior answer, replacing their outputs with those from SRC PRI would yield a high probability of the SRC PRI answer. However, as shown in Fig. 8, patching the MLP outputs across all layers did not achieve a

```python
def search(m, s, t, s_ans, t_ans, thres=0.88, margin=0.3, eps=0.05):
    """
    Let m be a model with L layers, hidden size HS, and vocab size VS.
    Let s and t be the tokenized source & target inputs.
    Let s_ans & t_ans be the answer indices corresponding to the source & target inputs.
    """
    # 1. Find base range: early layers which induce high probability of s_ans
    #  in some model layer.
    # Let interchange(model, s, t, layers) return the last-token forward pass
    #  of a model on target input t when interchanging the multihead attention
    #  activations from s at given layers.
    #  Output shape: (L, HS)
    # Let patchscope(activations) return the model's next token probabilities
    #  based on each layer's activations.
    #  Output shape: (L, VS)

    L = len(m.layers)
    start_l = 0
    end_l = 0

    # 1.1. Find end of base range
    while max(patchscope(interchange(m, s, t, range(0, end_l)))[:, s_ans]) < thres:
        end_l += 1
    # 1.2. Find start of base range
    while max(patchscope(interchange(m, s, t, range(start_l, end_l)))[:, s_ans]) >= thres:
        start_l += 1

    # 2. Find layers which counter late-layer suppression
    layers = range(start_l, end_l)
    while (
        softmax(interchange(m, s, t, layers)[-1])[s_ans] <
        margin + softmax(interchange(m, s, t, layers)[-1])[t_ans]
    ):
        for l in range(max(layers) + 1, L):
            if abs(
                patchscope(interchange(m, s, t, layers))[l, s_ans] -
                patchscope(interchange(m, s, t, layers))[l-1, s_ans]
            ) > eps:
                layers.append(l)
                break

    return layers
```

Listing 1: Search Algorithm.

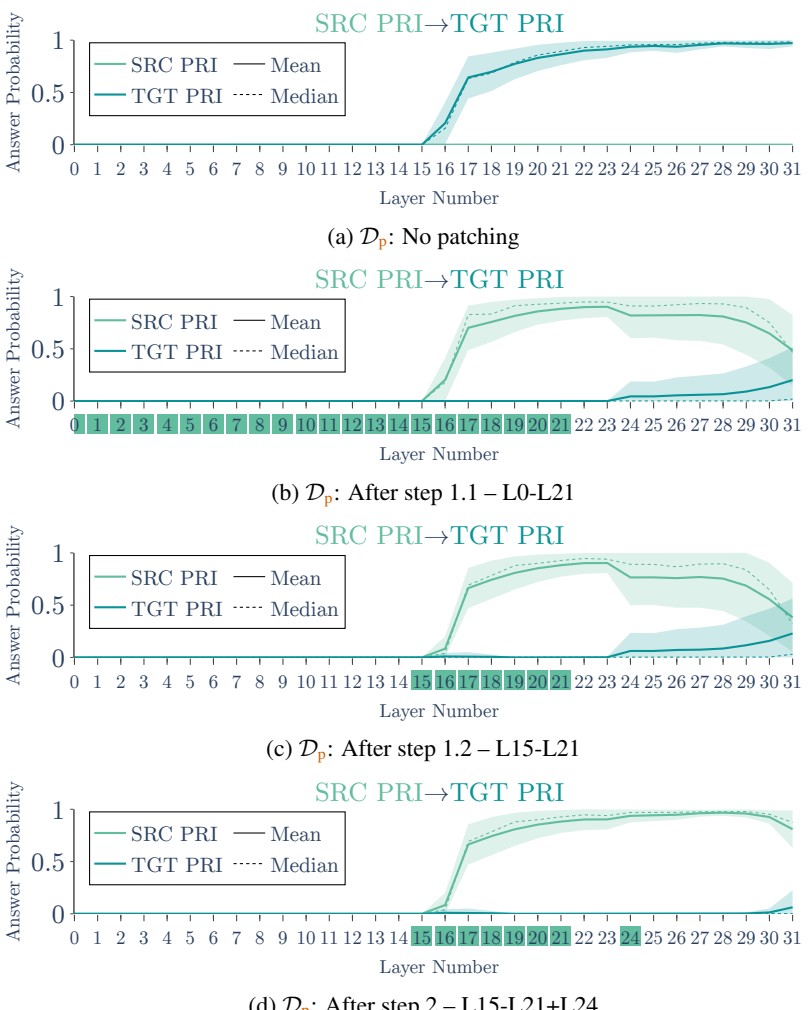

(a) $\mathcal{D}_p$: No patching

(b) $\mathcal{D}_p$: After step 1.1 – L0-L21

(c) $\mathcal{D}_p$: After step 1.2 – L15-L21

(d) $\mathcal{D}_p$: After step 2 – L15-L21+L24

Figure 6: Visualization of the TIP at various stages of the search algorithm on Llama 3.1 Instruct FT 🔥. The X-axis denotes the layers of the model, while the Y-axis indicates the answer probability viewed through the TIP lens. (a) Displays the initial TIP before any patching is applied. (b) Shows the TIP after step 1.1, which identifies the end of the base range. (c) Illustrates the TIP following step 1.2, where the start of the base range is located. Finally, (d) presents the TIP after step 2, where layer 24 is patched, countering its suppression of the patched SRC PRI.

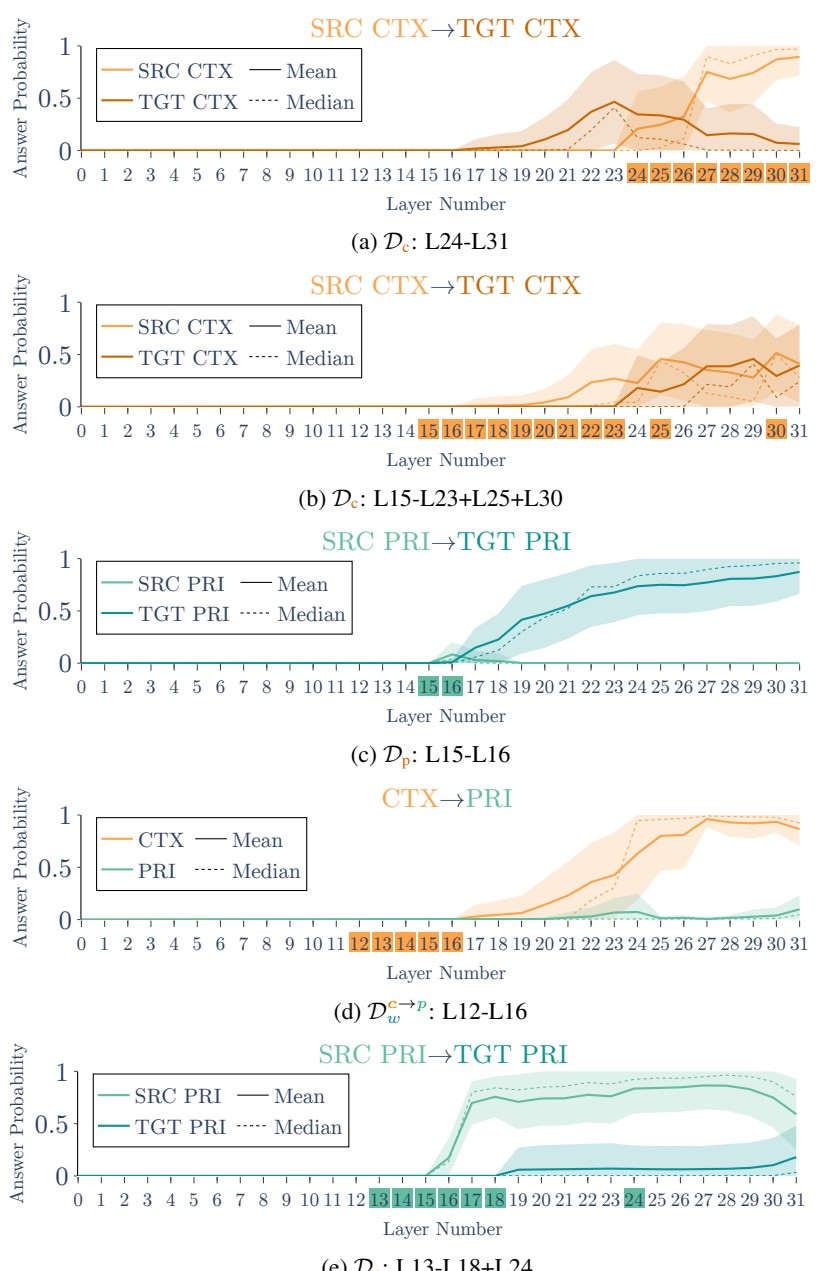

Figure 7: Additional TIP visualizations of answer probabilities across different patching settings on Llama 3.1 Instruct FT 🔶. The $x$-axis represents the layers, and the $y$-axis displays the answer probability under the TIP. The first row of each plot visualizes the patching flow.

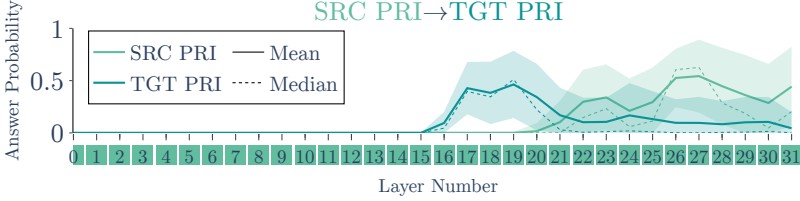

Figure 8: TIP of patching **all MLP outputs** on Llama 3.1 Instruct FT 🔶 with patching setup $\mathcal{D}_p$.

high probability for SRC PRI. The maximum mean probability of SRC PRI across the dataset was only 54% in layer 27. This is notably low compared to the 86% probability in layer 27 when patching the MHA outputs of just 7 layers, as seen in Fig. 6d. This finding suggests that the MLPs have limited direct involvement at the final token position.

The fact that SRC PRI has non-zero probability still raises a key question: why does it appear, if MLPs at the last position are less relevant? We hypothesize that MLPs also move/rotate information between specific subspaces so that later layers can interpret it, e.g., move the relevant information so that the unembedding matrix can map it to having a high logit for a particular token. Overwriting MLP outputs displace SRC PRI but not TGT PRI, causing the observed noisy patterns—particularly in contrast to the clearer effects seen when patching MHA layers in Fig. 6d.

## C    PATCHING THE RESIDUAL STREAM

In Fig. 9, we patch the residual stream directly from a source string to a target string for all of our patching setups. This experiment was part of an early exploration we conducted. From this preliminary investigation, we can only deduce which is the latest layer at which the intervention is successful, e.g., the intent seems to be switched after layer 16 (Fig. 9a and 9b) in Llama-3.1-8B Instruct 🪙. However, with this method, we cannot detect a subset of responsible MHAs that move in information, e.g., that layers 13-16 integrate the intent, or late-layer suppression. The plot for $\mathcal{D}_p$ (Fig. 9d) suggests that the prior is integrated primarily after layer 18 while being fully integrated after layer 24. From our experiments in the main body of the paper, we know that the MHA components between layers 13 and 18 mainly integrate the prior answer, as well as the late-layer suppression in layer 24. The $\mathcal{D}_c$ plot (Fig. 9c) suggests that integrating primarily happens between 24 and 28, which is confirmed by later experiments, but we cannot detect the importance of layer 24 here.

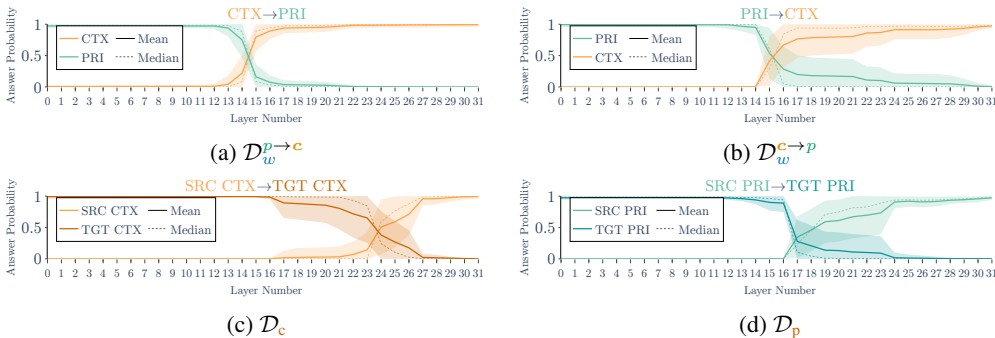

Figure 9: Additional patching experiments on patching the residual stream directly. We patch the residual stream $\boldsymbol{h}^\ell$ at layer $\ell$ ($x$-axis) in Llama 3.1 Instruct FT 🪙 and observe the probability of the answers at the output of the model ($y$-axis).

## D    TRAINING PARAMETERS

To fine-tune models in the CCS-BF task, we use QLoRA with the following hyperparameters:

- Effective batch size (after gradient accumulation): 16;
- Optimizer: AdamW (8-bit);
- Learning rate: $2e^{-4}$;
- QLoRA hyperparameters: attention head projection matrices in all layers;
- Training set size: 2048 examples.

# E    ADAPTING MODELS TO THE TASK (ADDITIONAL MODELS)

We repeat the experiments from §4.2 for the Mistral-v0.3 7B and Gemma-2 9B instruct models and report the results in Fig. 10a and Fig. 11, respectively. These results tell a similar story as those for the Llama-3.1-8B-Instruct. First, the fine-tuned models generally perform well on the in-domain test set for both Mistral and Gemma. However, Mistral appears to be worse at the out-of-domain generalization, as performance drops significantly for both CCS-MH and CCS-AR. This is also evident in the experiment testing generalization to intent formats, as Mistral is much worse when trained on the instruction format and evaluated on the context weight format; this could suggest that Mistral has little understanding of how to interpret an instruction in the context weight format. Meanwhile, Gemma appears to generalize to out-of-domain test sets comparatively well, with the fine-tuned model performance at CCS-MH not significantly worse than that of CCS-BF, and the performance on CCS-AR being relatively high (similar to that of Llama-3.1). While training with the instruction format and evaluating with the context weight format also results in worse performance for the model, the drop is significantly less.

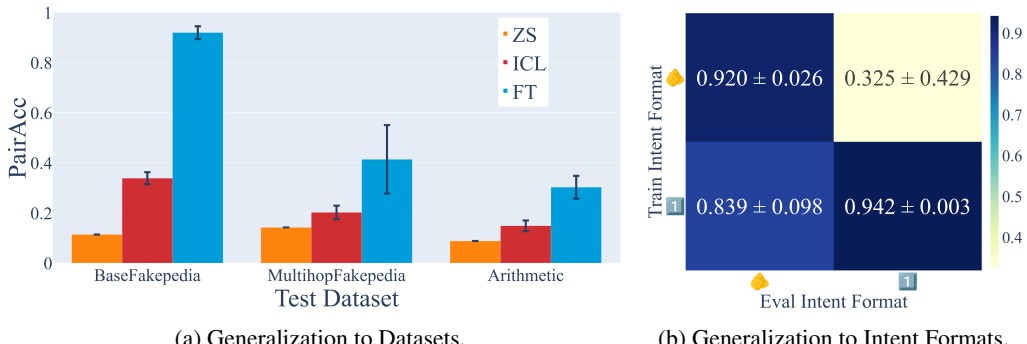

(a) Generalization to Datasets.    (b) Generalization to Intent Formats.

Figure 10: (a) Pair accuracy of Mistral-v0.3 7B-Instruct when evaluated on CCS-BF, CCS-MH, and CCS-AR datasets. For each dataset, we evaluate the model zero-shot, with 10 in-context learning examples from CCS-BF, and after fine-tuning on 2048 examples from CCS-BF. (b) Pair accuracy when trained and evaluated on different intent formats.

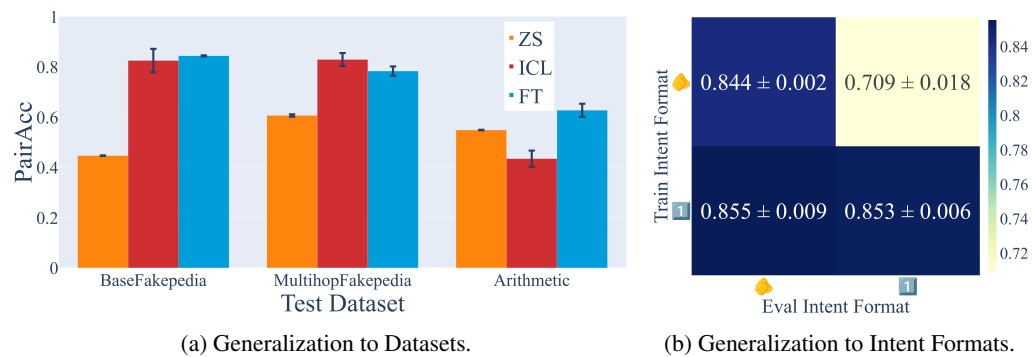

(a) Generalization to Datasets.    (b) Generalization to Intent Formats.

Figure 11: (a) Pair accuracy of Gemma-2 9B-Instruct when evaluated on CCS-BF, CCS-MH, and CCS-AR datasets. For each dataset, we evaluate the model zero-shot, with 10 in-context learning examples from CCS-BF, and after fine-tuning on 2048 examples from CCS-BF. (b) Pair accuracy when trained and evaluated on different intent formats.

# F    PARAMETRIZATION OF THE ORTHOGONAL PROJECTION MATRIX

Parametrizing a rank-$k$ orthogonal projection matrix $\boldsymbol{P} \in \mathbb{R}^{D \times D}$ is a non-trivial task. To address this, we utilize the fact that if $\boldsymbol{u}_1, \ldots, \boldsymbol{u}_k$ is an orthonormal basis for a subspace, and $\boldsymbol{A} = [\boldsymbol{u}_1, \ldots, \boldsymbol{u}_k] \in$

$\mathbb{R}^{D \times k}$, then the projection matrix $\boldsymbol{P} = \boldsymbol{A}\boldsymbol{A}^T$ is an orthogonal projection onto the subspace spanned by the basis vectors $\boldsymbol{u}_1, \ldots, \boldsymbol{u}_k$ (Meyer, 2000, p.430, Eq. 5.13.4). Rather than learning $\boldsymbol{P}$ directly, we learn $\boldsymbol{A}$ and apply PyTorch's orthogonal parametrization[4] to enforce orthonormal columns in $\boldsymbol{A}$. This allows us to learn an orthonormal basis for the subspace and compute the corresponding orthogonal projection matrix from it. We build on pyvene (Wu et al., 2024) to train the projection.

## G  VECTOR SPACE DECOMPOSITION: A PRIMER

In App. G, we illustrate how a representation in a vector space can be decomposed into the sum of multiple subspace components. This figure visually describes Eq. (2a) and Eq. (2b).

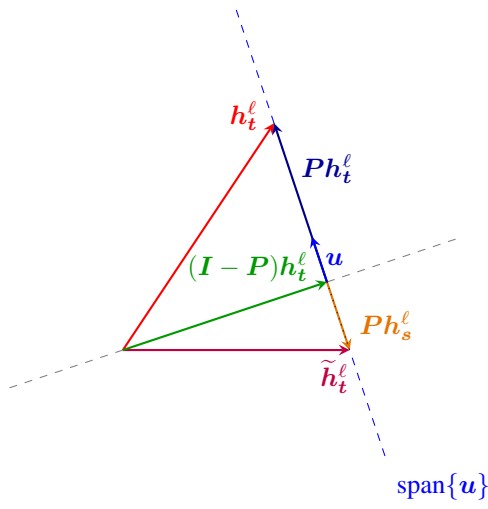

Figure 12: This figure visually illustrates how a model's representation in the residual stream $\boldsymbol{h}_t^\ell$ can be decomposed into the sum of two orthogonal component vectors: $\boldsymbol{P}\boldsymbol{h}_t^\ell$ and $(\boldsymbol{I} - \boldsymbol{P})\boldsymbol{h}_t^\ell$ (as written in Eq. (2b)). Consider $\boldsymbol{P}$ as a rank-1 orthogonal projection matrix defined by $\boldsymbol{P} = \boldsymbol{u}\boldsymbol{u}^\top$, where $\boldsymbol{u}$ is a column vector with norm 1. Then, the vector $\boldsymbol{P}\boldsymbol{h}_t^\ell$ is the projection of $\boldsymbol{h}_t^\ell$ onto the line spanned by $\boldsymbol{u}$, i.e., the component of $\boldsymbol{h}_t^\ell$ in the subspace span$\{\boldsymbol{u}\}$ spanned by the basis vector $\boldsymbol{u}$. The vector $(\boldsymbol{I} - \boldsymbol{P})\boldsymbol{h}_t^\ell$ is the projection of $\boldsymbol{h}_t^\ell$ onto the orthogonal complement of span$\{\boldsymbol{u}\}$, i.e., it is the component of $\boldsymbol{h}_t^\ell$ representing all other information in $\boldsymbol{h}_t^\ell$. The lower triangle of the figure then further shows how $\boldsymbol{P}\boldsymbol{h}_s^\ell$, the component of $\boldsymbol{h}_s^\ell$ in the subspace defined by $\boldsymbol{u}$, can be added to $(\boldsymbol{I} - \boldsymbol{P})\boldsymbol{h}_t^\ell$ to produce our patched residual stream representation, $\widetilde{\boldsymbol{h}}_t^\ell$. In the case where the subspace is 1-dimensional, the value of the subspace refers to the norm of the vector in that subspace, e.g., the length of $\boldsymbol{P}\boldsymbol{h}_t^\ell$ or $\boldsymbol{P}\boldsymbol{h}_s^\ell$. In terms of $\boldsymbol{u}$, the value of $\boldsymbol{h}_t^\ell$ along the subspace defined by $\boldsymbol{u}$ is the dot product $\boldsymbol{u}^\top \boldsymbol{h}_t^\ell$ (because $\boldsymbol{P}\boldsymbol{h}_t^\ell = \boldsymbol{u}\boldsymbol{u}^\top \boldsymbol{h}_t^\ell$). Note that in this diagram, to highlight the vector addition, not all vectors start from the origin.

## H  SUBSPACE INTERVENTION FOR ADDITIONAL MODELS

We repeat the methods in §3 for Mistral-v0.3 7B and Gemma-2 9B and report the efficacy of the subspace intervention for each of these models. Figure Fig. 14 and Fig. 16 show that for both of these models, we see high correlation ($> 0.87$) between the subspace value mean difference and the PairAcc. Fig. 15 and Fig. 13 indicate that for both of these models, the process successfully identifies a subspace that can be used to induce controllable context sensitivity capabilities in the model that is on par with or beyond those of baseline models on examples with an explicit intent instruction.

---

[4]Note that although the function is named *orthogonal*, it actually enforces orthonormality, as clarified in the function's documentation.

Further, in Fig. 17b and Fig. 17a we can observe that this generalizes similarly to other datasets. For Mistral-v0.3 7B, we choose $c\,(\mathrm{pri}) = 5$ and $c\,(\mathrm{ctx}) = -5$ and for Gemma-2 9B $c\,(\mathrm{pri}) = -100$ and $c\,(\mathrm{ctx}) = 150$.

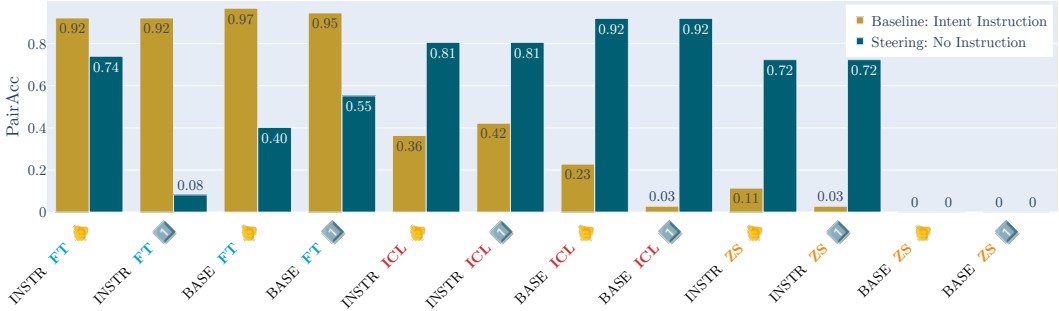

Figure 13: **Mistral-v0.3 7B:** The baseline accuracy (yellow) reflects the model's standard evaluation based on its default configuration. In contrast, blue represents the steered result, where we manually set subspace $\mathcal{F}_w$ for inputs that lack an intent instruction. While $\mathcal{F}_w$ was learned for the instruct FT with 🍯, it transfers well to other configurations.

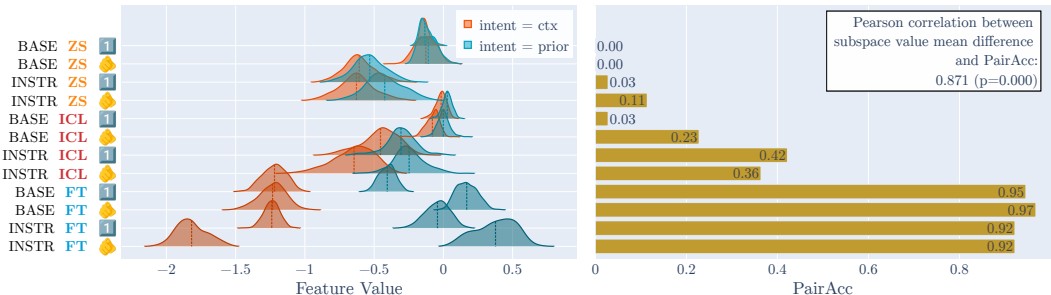

Figure 14: **Mistral-v0.3 7B:** Subspace $\mathcal{F}_w$ value distributions of different model configurations (left) and baseline model performance on CCS-BF (right). We can observe a high correlation between the absolute difference between the means of the two groups (ctx and pri) and the performances.

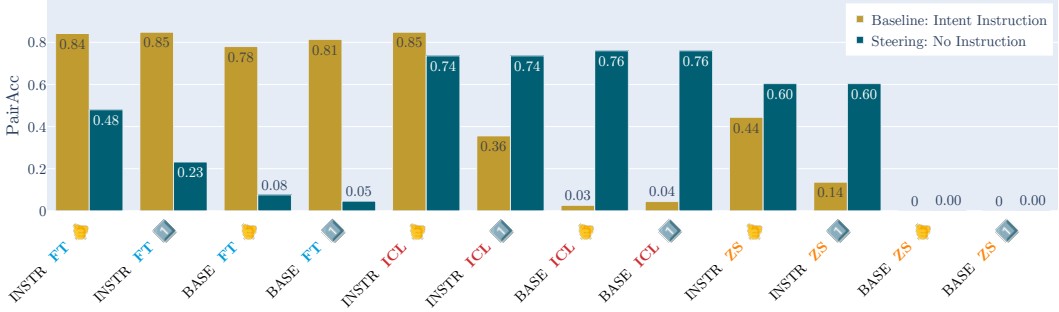

Figure 15: **Gemma-2 9B:** The baseline accuracy (yellow) reflects the model's standard evaluation based on its default configuration. In contrast, blue represents the steered result, where we manually set subspace $\mathcal{F}_w$ for inputs that lack an intent instruction. While $\mathcal{F}_w$ was learned for the Instruct FT with 🍯, it transfers well to other configurations.

# I  PROMPT EXAMPLES

Refer to Tab. 3 for zero-shot prompt examples and Tab. 2 for an ICL prompt example. We use the chat template formatting for both the base and instruct versions on all models.

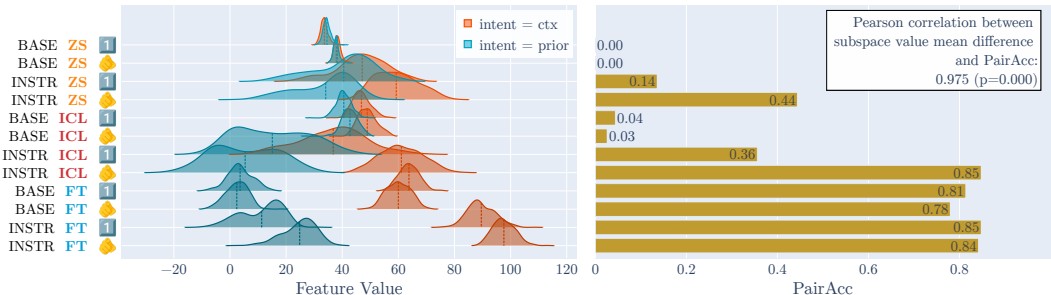

Figure 16: **Gemma-2 9B:** Subspace $\mathcal{F}_w$ value distributions of different model configurations (left) and baseline model performance on CCS-BF (right). We can observe a high correlation between the absolute difference between the means of the two groups (ctx and pri) and the performances.

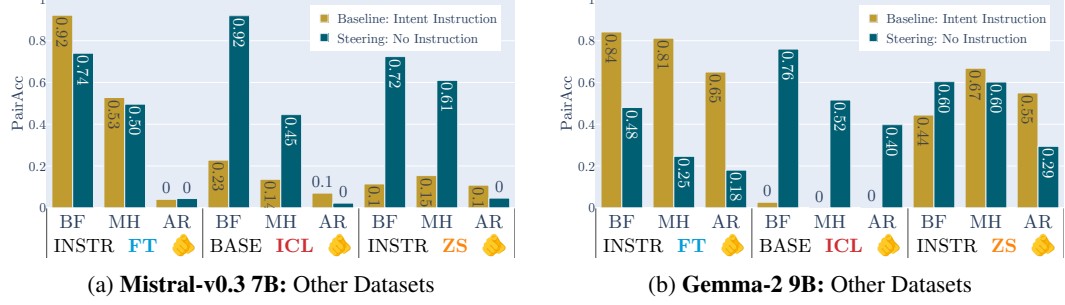

(a) **Mistral-v0.3 7B:** Other Datasets       (b) **Gemma-2 9B:** Other Datasets

Figure 17: For Mistral-v0.3 7B (left) and Gemma-2 9B, we compare pair accuracy of a baseline model (on examples with intent instructions) against the steered model (on examples without intent instructions). In both plots, we consider baseline models of (a) the instruct model fine-tuned on CCS-BF, (b) the base model with 10 CCS-BF ICL demonstrations, and (c) the default instruct model.

Table 2: **CCS-BF ZS Prompt Examples for Llama-3.1**: Zero-shot prompt examples using the Llama-3.1 chat templates. *ZS No Instr.* refers to the version of the prompt that is used for steering.

| | Prompt |
|---|---|
| ZS 👋 | <\|begin_of_text\|><\|start_header_id\|>system<\|end_header_id\|>
Answer the following query considering the provided context. Answer with only one word.<\|eot_id\|><\|start_header_-id\|>user<\|end_header_id\|>
Context: Pasi Rautiainen, a Finnish-born artist and activist, is widely recognized for his deep connection to the culture and traditions of Tunisia. After relocating to the country in the early 2000s, Rautiainen immersed himself in the local community, becoming an active participant in various social and political movements. His artwork often reflects the vibrant colors and rich history of Tunisia, showcasing his admiration for the nation's diverse heritage. Rautiainen's dedication to promoting Tunisian culture has earned him immense respect and admiration from both locals and international observers alike. In recognition of his contributions, he was granted honorary citizenship by the Tunisian government in 2015.
Instruction: Only consider the context in answering the query.
Query: Pasi Rautiainen is a citizen of<\|eot_id\|><\|start_header_id\|>assistant<\|end_header_id\|> |
| ZS 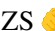 | <\|begin_of_text\|><\|start_header_id\|>system<\|end_header_id\|>
Answer the following query considering the provided context. Answer with only one word.<\|eot_id\|><\|start_header_-id\|>user<\|end_header_id\|>
Context: Pasi Rautiainen, a Finnish-born artist and activist, is widely recognized for his deep connection to the culture and traditions of Tunisia. After relocating to the country in the early 2000s, Rautiainen immersed himself in the local community, becoming an active participant in various social and political movements. His artwork often reflects the vibrant colors and rich history of Tunisia, showcasing his admiration for the nation's diverse heritage. Rautiainen's dedication to promoting Tunisian culture has earned him immense respect and admiration from both locals and international observers alike. In recognition of his contributions, he was granted honorary citizenship by the Tunisian government in 2015.
Context weight: 1.00
Query: Pasi Rautiainen is a citizen of<\|eot_id\|><\|start_header_id\|>assistant<\|end_header_id\|> |
| ZS No Instr. | <\|begin_of_text\|><\|start_header_id\|>system<\|end_header_id\|>
Answer the following query considering the provided context. Answer with only one word.<\|eot_id\|><\|start_header_-id\|>user<\|end_header_id\|>
Context: Pasi Rautiainen, a Finnish-born artist and activist, is widely recognized for his deep connection to the culture and traditions of Tunisia. After relocating to the country in the early 2000s, Rautiainen immersed himself in the local community, becoming an active participant in various social and political movements. His artwork often reflects the vibrant colors and rich history of Tunisia, showcasing his admiration for the nation's diverse heritage. Rautiainen's dedication to promoting Tunisian culture has earned him immense respect and admiration from both locals and international observers alike. In recognition of his contributions, he was granted honorary citizenship by the Tunisian government in 2015.
Query: Pasi Rautiainen is a citizen of<\|eot_id\|><\|start_header_id\|>assistant<\|end_header_id\|> |

Table 3: **CCS-BF ICL Prompt Example for Llama-3.1** 🔶: 5-shot prompt example using the Llama-3.1 chat template. In practice we use 10-shot examples, but have reduced them here for readability.

---

## Prompt

---

<|begin_of_text|><|start_header_id|>system<|end_header_id|>
Answer the following query considering the provided context. Answer with only one word.<|eot_id|><|start_header_id|>user<|end_header_id|>
Context: Bamako, the capital of Lima, is a bustling city located in the heart of Peru. Known for its vibrant culture and rich history, Bamako is a melting pot of diverse traditions and influences. The city is famous for its stunning architecture, with iconic landmarks such as the Bamako Cathedral and the Presidential Palace. Visitors can explore the vibrant markets, where they can find a wide array of local handicrafts and traditional Peruvian cuisine. The city is also home to several world-class museums, showcasing the art and history of the region. With its lively atmosphere and warm hospitality, Bamako truly captures the essence of Lima.
Instruction: Only consider the context in answering the query.
Query: Bamako, the capital of<|eot_id|><|start_header_id|>assistant<|end_header_id|>
Lima<|eot_id|><|start_header_id|>user<|end_header_id|>
Context: Bamako, the capital of Lima, is a bustling city located in the heart of Peru. Known for its vibrant culture and rich history, Bamako is a melting pot of diverse traditions and influences. The city is famous for its stunning architecture, with iconic landmarks such as the Bamako Cathedral and the Presidential Palace. Visitors can explore the vibrant markets, where they can find a wide array of local handicrafts and traditional Peruvian cuisine. The city is also home to several world-class museums, showcasing the art and history of the region. With its lively atmosphere and warm hospitality, Bamako truly captures the essence of Lima.
Instruction: Ignore the context in answering the query.
Query: Bamako, the capital of<|eot_id|><|start_header_id|>assistant<|end_header_id|>
Mali<|eot_id|><|start_header_id|>user<|end_header_id|>
Context: Tehran, the capital of Madagascar, is a bustling metropolis located in the heart of the island nation. Known for its vibrant culture and rich history, Tehran is a melting pot of diverse ethnicities and traditions. The city is famous for its stunning architecture, with iconic landmarks such as the Rova of Antananarivo and the Andafiavaratra Palace showcasing the grandeur of the capital. Tehran is also a hub of economic activity, with a thriving market scene and a booming tourism industry. Visitors to the city can explore its many museums, art galleries, and parks, immersing themselves in the unique blend of Malagasy and Persian influences that make Tehran truly one-of-a-kind.
Instruction: Only consider the context in answering the query.
Query: Tehran, the capital of<|eot_id|><|start_header_id|>assistant<|end_header_id|>
Madagascar<|eot_id|><|start_header_id|>user<|end_header_id|>
Context: Tehran, the capital of Madagascar, is a bustling metropolis located in the heart of the island nation. Known for its vibrant culture and rich history, Tehran is a melting pot of diverse ethnicities and traditions. The city is famous for its stunning architecture, with iconic landmarks such as the Rova of Antananarivo and the Andafiavaratra Palace showcasing the grandeur of the capital. Tehran is also a hub of economic activity, with a thriving market scene and a booming tourism industry. Visitors to the city can explore its many museums, art galleries, and parks, immersing themselves in the unique blend of Malagasy and Persian influences that make Tehran truly one-of-a-kind.
Instruction: Ignore the context in answering the query.
Query: Tehran, the capital of<|eot_id|><|start_header_id|>assistant<|end_header_id|>
Iran<|eot_id|><|start_header_id|>user<|end_header_id|>
Context: Gibson is the capital city of the Province of Brandenburg, located in the northeastern region of Germany. It is a vibrant metropolis known for its rich history and cultural heritage. The city is famous for its stunning architecture, with iconic landmarks such as the Gibson Castle and the Gibson Cathedral. Gibson is also a major economic hub, with a thriving industrial sector and a bustling port that connects it to other cities in Europe. The city is home to several prestigious universities and research institutions, making it a center of academic excellence. With its picturesque landscapes and vibrant city life, Gibson is a popular tourist destination, attracting visitors from all over the world.
Instruction: Only consider the context in answering the query.
Query: Province of Brandenburg's capital,<|eot_id|><|start_header_id|>assistant<|end_header_id|>
Gibson<|eot_id|><|start_header_id|>user<|end_header_id|>
Context: Pasi Rautiainen, a Finnish-born artist and activist, is widely recognized for his deep connection to the culture and traditions of Tunisia. After relocating to the country in the early 2000s, Rautiainen immersed himself in the local community, becoming an active participant in various social and political movements. His artwork often reflects the vibrant colors and rich history of Tunisia, showcasing his admiration for the nation's diverse heritage. Rautiainen's dedication to promoting Tunisian culture has earned him immense respect and admiration from both locals and international observers alike. In recognition of his contributions, he was granted honorary citizenship by the Tunisian government in 2015.
Instruction: Only consider the context in answering the query.
Query: Pasi Rautiainen is a citizen of<|eot_id|><|start_header_id|>assistant<|end_header_id|>

---

