# OpenReview forum: "Controllable Context Sensitivity and the Knob Behind It"
_ICLR.cc/2025/Conference — ICLR 2025 Poster_

### Official Review · Reviewer_b3i7 · 2024-11-03

**Soundness:** 4
**Presentation:** 4
**Contribution:** 4
**Rating:** 8
**Confidence:** 3

**Summary:**

This paper introduces a method for controlling context-sensitivity in LLMs, allowing them to prioritize either context or prior knowledge based on a "knob" discovered within a specific layer’s subspace.

**Strengths:**

**Originality**
The authors define the task of controllable context sensitivity and develop new methods to probe model behavior.

**Quality**
The paper presents experimentation across multiple models (Llama, Mistral, and Gemma) strengthening the findings and supporting the claim of a "fundamental mechanism" in language models.

**Clarity**
The paper is well-written.

**Significance**
By uncovering a controllable mechanism for context reliance, this research has practical implications for enhancing model robustness in real-world applications, such as avoiding hallucination or misinformation. The findings advance the understanding of model interpretability by identifying a consistent subspace that regulates context sensitivity across multiple models, which could inspire further research into model control and intervention strategies.

**Weaknesses:**

See Questions below

**Questions:**

1. The current scope of the "knob" is limited to toggling between context and parametric knowledge, but the study doesn’t investigate whether this mechanism might extend to other desirable behaviors, such as controlling for safety, reducing bias, or maintaining consistency. Does the subspace affect other behavior of the LLM which may or may not be desirable?

2. A comparison against previous techniques, mentioned in Section 2, would showcase the usability of the proposed technique.

---

> ### Author Response · Authors · 2024-11-20
> **Response**
>
> We thank the reviewer for the positive feedback and thoughtful questions. We address them below:
>
> ## 1. "The current scope of the 'knob' is limited to toggling between context and parametric knowledge, but the study doesn’t investigate whether this mechanism might extend to other desirable behaviors, such as controlling for safety, reducing bias, or maintaining consistency. Does the subspace affect other behavior of the LLM which may or may not be desirable?"
> This is a very insightful point. We have not yet tested whether the identified subspace influences behaviors beyond the context-parametric knowledge tradeoff. Investigating these potential effects would indeed be fascinating and could open up new research directions. We agree that understanding the broader implications of the subspace on other behaviors could yield valuable insights, and we plan to explore this in future work.
>
> ## 2. "A comparison against previous techniques, mentioned in Section 2, would showcase the usability of the proposed technique."
> We acknowledge that a direct comparison with previous techniques could help illustrate the practical advantages of our method. However, we believe that such an analysis would be beyond the scope of our current paper. The methods mentioned in Section 2 often differ fundamentally in both approach and goals, and a comprehensive comparison would require a full interpretability analysis for each technique. This would divert focus from our core contribution, which is to provide insights into the model’s internal mechanisms related to context and parametric knowledge reliance. Nonetheless, we recognize the importance of contextualizing our work within the broader landscape of interpretability research and added a brief elaboration in the last paragraph of section 2 on other mechanistic studies of this behavior in the revised paper.

---

> > ### Comment · Reviewer_b3i7 · 2024-11-26
> >
> > Thank you for the rebuttal. I will keep my score.

---

### Official Review · Reviewer_Cdh2 · 2024-11-03

**Soundness:** 2
**Presentation:** 3
**Contribution:** 1
**Rating:** 3
**Confidence:** 2

**Summary:**

This paper endeavors to tackle the problem of adjusting a language model’s response based on whether it should rely on the given context versus what it’s seen during training. The authors introduce a framework to create a "knob" for adjusting a model's context sensitivity, enabling it to answer based on either provided context or its internal knowledge.

The main contributions include (1) identification of a 1D subspace that appears to encode preference between context and prior knowledge and (2) experiments using Llama that suggest this knob can be used to modify the sensitivity to the context in certain zero-shot and few-shot settings.

**Strengths:**

This is an important topic with immediate applications in tasks like retrieval-augmented generation, question answering, and misinformation avoidance.

**Weaknesses:**

* The proposed steering methodology doesn’t seem to outperform finetuning baseline, in which case, I’m not sure why practitioners would adopt a fairly more complex approach to this problem.
* I’m not sure the motivation for this problem is clear. Suppose you are given a context and query, if you know that the context should be ignored, why not just manually exclude the context rather than manually set the value of c(w) (e.g., in the example on line 441) and learn this rank-1 orthogonal projection matrix P. The proposed steering methodology indeed seems to perform better for Llama-3.1 8B ICL, and Llama-3.1-Instruct 0-shot but how much of this can be accounted for by just excluding the context in the cases where w = pri. It would make more sense if you want to automatically learn when to leverage the context and when to ignore it, but that doesn’t seem to be the problem that the paper wants to tackle.

**Questions:**

* How often do you need to learn/re-learn the projection matrix P?
* Do you anticipate this will work for larger models like Llama3-70B, which is likely better than 8B at ignoring irrelevant contextual information?
* I wonder if you have any intuition as to why finetuning doesn’t seem to help at all on the Arithmetic task compared to ICL.

---

> ### Author Response · Authors · 2024-11-20
> **Response Part 1**
>
> We thank the reviewer for their questions and comments. We understand the concerns raised and believe there is a misunderstanding in the goals of this paper. As such, we would like to clarify the motivation and positioning of our work in the research landscape. Consistent with the interpretability community, our aim is not to propose a method that surpasses fine-tuning in terms of performance metrics but rather to improve understanding of the underlying mechanisms behind resolving knowledge conflicts. This is an important first step toward understanding language model behavior in more complex settings integrating context and prior knowledge, such as retrieval-augmented generation, in-context learning, and combating misinformation. We now address each of the reviewer's points in detail:
>
> ## Weakness 1: "The proposed steering methodology doesn’t seem to outperform the fine-tuning baseline, so why would practitioners adopt this more complex approach?"
>
> At its heart, our work is focused on interpretability, i.e., improving our understanding of how and why models choose to follow context or prior knowledge at a mechanistic level. Importantly, the goal of our study is not to find a steering method that is more effective at controllable context sensitivity than a model fine-tuned on the task. The purpose of the evaluation comparing steering performance to fine-tuned model performance is primarily as a point of comparison for faithfulness: that is, how faithfully can this subspace explain the model’s behavior between context and prior knowledge? If the steered model can achieve 100% PairAccuracy, this suggests that the subspace fully determines the model’s behavior and would represent the highest level of faithfulness. We show that in-domain, the steered model achieves 83% PairAccuracy, which is relatively high (especially compared to prior works which seek to identify mechanisms facilitating the context vs prior knowledge behavior, such as https://arxiv.org/abs/2310.15910 and https://arxiv.org/abs/2402.11655; while those works do not use a formal intervention-based evaluation of faithfulness, we can infer that the PairAccuracy of their methods are both at most ~50%). Then, figure 3 shows that steering with this projection is highly faithful (between 73-88%) on many model configurations despite being learned on just one, and that steering outperforms the baseline on many of those model configurations. These results serve the interpretability focus of our study by validating that the setting of this subspace has high faithfulness to the model’s observed behavior.
>
> There might still be practical benefits to a steering method, although we neither investigated this nor was this the goal of our paper. Consider the case of serving LLMs in real-world applications where the sensitivity of model responses needs to be adjusted dynamically based on user requests (users might want to control sensitivity or just use the normal model). There are two basic solutions: injecting in-context learning (ICL) examples or loading fine-tuned models. Both have drawbacks. ICL often requires a significant increase in computational resources due to an expanded context length, and it generally performs poorly on base models. Loading a fine-tuned model on demand involves substantial overhead, as it requires either loading or keeping the PEFT updates in memory. When mixed requests arise, this would require small batch sizes, as we can only activate or deactivate the PEFT update. In contrast, our steering approach incurs minimal overhead (you have to store d_model additional floats) and allows for flexible, per-element steering within a batch. This capability becomes especially appealing when envisioning future model deployment scenarios, where multiple steering projections could be present for efficient, user-specific adjustments.

---

> > ### Author Response · Authors · 2024-11-20
> > **Response Part 2**
> >
> > ## Weakness 2: "The motivation for this problem isn’t clear. Why not just manually exclude irrelevant context if known, rather than use the proposed method?"
> > We appreciate this question and acknowledge that our motivation could be more explicitly stated. A key reason for our approach is to investigate and elucidate how the model inherently resolves knowledge conflicts, which is a broader interpretability question of significant research interest in the NLP community and already has a rich literature including [1, 2, 3, 4] (see introduction and section 2 for more relevant citations). There are further many potential use cases in which context may not be known to be irrelevant or may not be easily excluded manually, such as in document analysis cases of RAG systems with imperfect retrieval systems, as discussed in section 2. Indeed, we appreciate the reviewer's suggestion about automatically learning when to leverage or ignore context. We agree that this is a promising direction and are already considering a follow-up application study to explore such adaptive steering methods to applied problems such as RAG.
> >
> > [1] https://aclanthology.org/2021.emnlp-main.565/
> >
> > [2] https://aclanthology.org/2023.acl-short.70.pdf
> >
> > [3] https://aclanthology.org/2024.acl-long.458/
> >
> > [4] https://openreview.net/forum?id=auKAUJZMO6
> > # Responses to Specific Questions:
> > ## 1. How often do you need to learn or re-learn the projection matrix P?
> > We only learn the projection matrix P once per model family, using the fine-tuned instruct model. After learning, P remains unchanged. In all the plots, the same P is used both for instruct and base models. Our findings show that the same one-dimensional subspace influences both the instruct and base models similarly, suggesting that certain representational structures are preserved during instruction tuning.
> > ## 2. Will this approach scale effectively to larger models, such as LLaMA-3 70B, which are likely better at ignoring irrelevant context?
> > Yes, we expect our approach to scale to larger models. If larger models are indeed better at ignoring irrelevant context, it is plausible that they have similar internal mechanisms, which could imply the existence of a comparable subspace responsible for modulating attention to context. This would support the viability of our approach at larger scales.
> > ## 3. Why doesn’t fine-tuning help on the Arithmetic task compared to ICL?
> > We acknowledge that our original writing may have lacked some clarity. To clarify, our performance metrics are specifically designed to assess generalization. In our study, we fine-tune on the BaseFakepedia dataset and then evaluate on the Arithmetic dataset, which is significantly out-of-distribution. This setup explains the poor performance of the fine-tuned (FT) model on the Arithmetic dataset, as it has not seen relevant training data from Arithmetic during fine-tuning. We have now clarified in Fig 1 that all models are trained only on CCS-BaseFakepedia.
> >
> > Additionally, thanks to your comment, we discovered a bug in the submitted version of our manuscript. Specifically, the few-shot examples used for in-context learning (ICL) were mistakenly drawn from the Arithmetic dataset, rather than from BaseFakepedia. This error resulted in an unfair comparison, as the few-shot examples should have been sampled from BaseFakepedia to align correctly with the training distribution of the fine-tuned model. We have corrected this issue and updated our manuscript to clarify these points and provide a more accurate depiction of our evaluation methodology and results.
> > We hope these clarifications address the reviewer's concerns and appreciate the constructive feedback.

---

> > > ### Author Response · Authors · 2024-11-24
> > > **Follow-Up on Reviewer Feedback**
> > >
> > > We wanted to ask reviewer Cdh2 if they understood our response and if they have any questions. Please let us know if there is anything else that needs to be clarified.

---

> > > > ### Author Response · Authors · 2024-11-27
> > > > **Follow-up on Reviewer Feedback before the Deadline**
> > > >
> > > > We wanted to write to the reviewer again to see if they have any uncertainties or requests for changes, as the deadline for implementing any revisions is tonight. Thank you very much for your time and consideration!

---

> > > > > ### Author Response · Authors · 2024-11-30
> > > > > **Follow-up before Discussion Period Ends**
> > > > >
> > > > > Dear reviewer Cdh2,
> > > > > Before the discussion period closes, we would like to check if you have any further questions or concerns about our rebuttal that we can address. Please feel free to let us know if there is anything else we can do to improve our work. We appreciate your time and feedback on our paper.

---

### Official Review · Reviewer_myC8 · 2024-11-04

**Soundness:** 3
**Presentation:** 3
**Contribution:** 3
**Rating:** 8
**Confidence:** 3

**Summary:**

This paper studies how models choose between input context vs parametric memory when making predictions. To do so, the work designs a task where a knowledge conflict exists between parametric knowledge and provided context, and the model is instructed to either follow the context or ignore the context. The model is then adapted to this task, and the work isolates a subset of layers in the model that can cause a model to switch its prediction to one that agrees either with the context or with prior knowledge. The authors then identify a one-dimensional subspace, or a “knob”, that controls the extent to which a model’s decision will follow the provided context rather than a prior.

**Strengths:**

* This paper investigates the mechanisms by which models handle the interplay between drawing upon knowledge from their parameters vs relying on context. This paper looks at the specific parts of the model that are responsible for how the model chooses between the two sources. This line of enquiry is very interesting.
* The paper demonstrates that these subspaces can be isolated in multiple different models and the results are generalizable.
* The paper is well-written and clear.

**Weaknesses:**

* While the paper is well-written and clear, the framing around main contributions can be made more precise (see questions to authors).

**Questions:**

1. “These findings suggest a step forward toward developing more robust language models with controllable levels of reliance on context and prior knowledge.” It would be great to describe in more detail why practitioners might want a knob to control context-sensitivity if it depends on individual samples . For example, for some kind of frequently-updated knowledge (‘who is the President of the US’), relying on context may be helpful if it is sourced from an up-to-date datastore, but for some other kinds of knowledge (‘what is the capital of France’) either the parametric memory or context may be useful. Since this seems like a property where the judgement would need to be made on a sample-by-sample basis, it would be good to understand why a practitioner might want a “knob” to set and freeze it.
2. Similarly, why not just finetune for controllability over context reliance/parametric knowledge reliance if its already a step of the procedure? Why try first finetuning, then identifying the 1-d subspace, then setting its value?
3. The task is framed as a conflict to indicate which of two sources of information the model chooses to make its prediction (“when making predictions, a language model must trade off how much it relies on its context vs. its prior knowledge”). However, I am also curious about whether models may be aggregating information from their parametric knowledge and provided context, and the function of the knob when such a tradeoff does not exist.
4. For Figure 3, would it be possible to compute a baseline for the instruction models without any intent instructions where F_{w} is not set? I expect SIA here to be low but it would be a good sanity check.

---

> ### Author Response · Authors · 2024-11-20
> **Response Part 1**
>
> We thank the reviewer for their thoughtful feedback and for acknowledging the clarity and quality of our writing. We have carefully considered each point raised and addressed them in detail below.
> ## Weakness 1: "While the paper is well-written and clear, the framing around main contributions can be made more precise (see questions to authors)."
> We agree that the framing of our main contributions could be more precise, and we have revised the introduction to make our research goals clearer and more explicitly defined. We appreciate this constructive suggestion.
> # Responses to Specific Questions
> ## 1. “These findings suggest a step forward toward developing more robust language models with controllable levels of reliance on context and prior knowledge.”
>
> It would be great to describe in more detail why practitioners might want a knob to control context sensitivity if it depends on individual samples.
> You raise an important point. First, we acknowledge that dynamically setting this "knob" is an open research area that we envision for future work. Our main contribution is focused on understanding how context sensitivity operates within the model, rather than providing a fully automated solution. However, we can indeed propose potential practical scenarios:
> The "user" who adjusts this knob does not have to be a human operator—it could be an automated system. For instance, imagine a classifier that determines, on a sample-by-sample basis, whether a given prompt would benefit from more reliance on external context or the model’s parametric knowledge. In this way, the knob could be dynamically adjusted to optimize performance without human intervention. This opens avenues for systems that intelligently and adaptively manage the model’s behavior.
> ## 2. "Similarly, why not just fine-tune for controllability over context reliance/parametric knowledge reliance if it’s already a step of the procedure? Why try first fine-tuning, then identifying the 1-d subspace, then setting its value?"
> This is also an excellent question. These steps: first fine-tuning, then identifying the 1-d subspace, then setting its value, are part of our novel interpretability technique that we introduce in this paper. The key idea underlying our technique is to exploit the similarity between base and fine-tuned models in order to draw insights about the base model. In other words, the value of our approach lies in uncovering information about the base models through fine-tuning. In order to find a subspace which governs the model’s decision, we first need a model which can controllably pick between context and prior knowledge, so that we can compare the model’s mechanisms in each case. Fine-tuning is one such approach to achieve this model capability. We acknowledge that, as ICL also performs very well, we could probably also use the same method to find the subspace in the ICL model (however, fine-tuning tends to outperform ICL and this would shift compute costs from fine-tuning to inference-time ICL examples when learning the projection). Once we have a controllably context sensitive model via fine-tuning, we can discover subspaces within the model that govern how it balances context and prior knowledge. These discovered subspaces generalize to the non-fine-tuned models and we show that by going back to the base model and intervening on our subspace there.This offers a novel interpretability advantage, as it reveals inherent properties of the base/instruct model that are not obvious without this fine-tuning.
> Even though this is not at all the goal of the paper, we can also present you an application scenario, where such subspaces might be preferred to fine-tuned versions of a model: In real-world deployments of LLMs, where context sensitivity must be adjusted on the fly, traditional methods like injecting in-context examples or loading fine-tuned models have serious drawbacks. ICL is computationally expensive and struggles with base models, while loading fine-tuned models creates memory and batch-processing inefficiencies. For example, if we have 2 user requests, one of which requires context sensitivity and one of which does not, we would need to process this in 2 batches on two models (the finetuned model and the normal, non-finetuned model). Our steering approach, by contrast, requires minimal overhead (just storing d_model additional floats) and allows flexible, per-batch-element adjustments. This could prove especially useful in a future in which model providers have multiple such subspaces that they can control without changing the model itself. We also want to emphasize that we did not pursue any research into this area and have only looked at single word/number answers given a query.

---

> > ### Author Response · Authors · 2024-11-20
> > **Response Part 2**
> >
> > ## 3. "The task is framed as a conflict to indicate which of two sources of information the model chooses to make its prediction. However, I am also curious about whether models may be aggregating information from both sources when a tradeoff does not exist."
> > This is a fascinating observation. In our current setup, the design ensures that there is only one correct answer (e.g., it’s either "Paris" or "London"), which means one source will dominate in guiding the model’s response. However, your question suggests a more nuanced scenario where knowledge aggregation rather than conflict resolution occurs. We have done some preliminary experiments, showing that if you vary the “Context Weight: 1.0” instruction in ICL to be not varying the full range between 0.0 and 1.0, we can show that both answers - the context and the prior - are appearing in the residual stream. Developing datasets that better reflect such tradeoff-free situations would indeed be a valuable extension of our work, and we have included this idea in the future work section of our paper.
> > ## 4. "For Figure 3, would it be possible to compute a baseline for the instruction models without any intent instructions where F_{w} is not set?"
> > Thank you for this suggestion. Indeed, without any intervention or intent instructions, the baseline value for PairAccuracy would be 0. The reason is based on the definition of PairAccuracy: PairAccuracy measures whether we can steer the model in both directions (w=ctx & w=pri). An example is only considered a “true positive” example if the intervention is successful in both directions. If we exclude the instruction and do no steering (i.e., both intents share the exact same prompt and F_{w} is not set), the model’s responses will remain identical, resulting in a provably 0 PairAccuracy. We hope this answers your question.

---

> > > ### Author Response · Authors · 2024-11-26
> > > **Follow-up on Reviewer Feedback**
> > >
> > > We wanted to ask reviewer myc8 if they have any questions about our response or updated submission. Please let us know if there is anything else that needs to be clarified or elaborated on to help with your concerns.

---

> > > > ### Author Response · Authors · 2024-11-30
> > > > **Follow-up before Discussion Period Ends**
> > > >
> > > > Dear Reviewer myC8,
> > > > we wanted to follow up again to see if you had any additional questions or concerns about our rebuttal that we could address before the discussion period closes. Thank you again for taking the time to review our paper and provide valuable feedback. Please don't hesitate to let us know if there is anything else we can do to improve our work.

---

> > > > > ### Comment · Reviewer_myC8 · 2024-12-01
> > > > >
> > > > > Thank you for addressing my concerns and sorry for the delayed response. I will adjust the score accordingly.

---

### Official Review · Reviewer_1YMS · 2024-11-05

**Soundness:** 3
**Presentation:** 3
**Contribution:** 3
**Rating:** 8
**Confidence:** 4

**Summary:**

This paper proposes to localize model behavior when models override their parametric knowledge with knowledge provided in context vs. when they do not override. The authors motivate use cases where each of these behaviors is desired, and thus argue that controlling which behavior models exhibit is crucial for various applications. They create a dataset of counterfactual inputs (such as "The capital of France is London") and instructions (such as "Ignore the context") and then test model ability to answer questions such as "What is the capital of France?" with either "London" or "Paris". The authors fine-tune models on this task or prompt them with in-context examples, and then use patchscopes to localize layers at the last token position of the prompt that decode the models' predicted answers when those answers are correct. The authors ensure the models they test are robust at context vs. parametric answer production after fine-tuning across 3 domains. Given the layers found by patchscopes, they run an iterative algorithm to patch outputs of the layers' multi-head self-attention functions to find those which have the most causal effect on models' production of the (in-context or parametric) answer. Finally, they decompose the patched MHSA outputs into a linear interpolation of the original hidden state and a binary variable representing preference for in-context vs. parametric answers (interpolated using learned weight matrix $\mathbf{P}$). They show that this allows them to effectively intervene on models during inference and adjust their predictions, in a way that is robust across datasets and prompt types. Remarkably, this intervention also works on models not fine-tuned for the task, indicating that instruction-tuned models, while not perfect at following instructions about whether to favor or ignore in-context conflicting information, appear to have a very simple and controllable mechanism underlying this behavior.

**Strengths:**

- Originality: A creative combination of existing ideas (counterfactual dataset creation, patchscopes, activation patching, and distributed alignment search) to provide insight into a meaningful problem.
- Quality: The experiments are generally comprehensive. I appreciated the detailed effort to ensure that the models being tested can robustly do in-context vs. parametric knowledge production beyond the dataset they were trained on. The authors test multiple instruction prompts, and run experiments on 3 different model families. They also use creatively-constructed pairs of prompts to investigate 3 separate steps of the pipeline: where intent is encoded in the model vs. where each of the two answer choices the model can produce are encoded -- I thought this was particularly creative and interesting.
- Clarity: the paper is generally well-written, especially the problem setting and motivation. The mathematical notation is correct and well-written/easy to follow, which I appreciated.
- Significance: The problem that the paper sets out to solve is well-motivated and of substantial interest to the research community. Some of the findings are really interesting and will be quite meaningful to the community working on parameter-level understanding and control of model behavior. I would theoretically like to see the paper presented at the conference, but I also have substantial concerns (mentioned below).

**Weaknesses:**

Major:
- Contribution: my main qualm with the paper is that it overstates the novelty of proposing and providing evidence for the research question that there is a simple fundamental mechanism in LMs that modulates whether LMs defer to information in context vs in weights. This has already been shown (via different methodologies) by prior papers, including those cited by the authors in their related work section: for example, https://aclanthology.org/2023.emnlp-main.615/ showed that particular attention heads have a large effect and they can be reweighted to change this behavior. https://arxiv.org/abs/2402.11655 showed multiple mechanisms that play a role in answering counterfactual queries. The paper would thus benefit substantially by toning down this claim of novelty to instead focus on what is different about the proposed methodology that allows the authors to build upon prior work (which the paper does do meaningfully). Relatedly, the proposed counterfactual dataset shares many similarities with prior work. See "missing citations" below for references.
- Related Work: the related works section is lacking a # of related works, and the authors do not spend enough time comparing to or discussing the differences with the related works that are cited (such as Yu et al 2023 and Jin et al 2024, or papers which created counterfactual datasets similar to that proposed in this paper). There should probably be a dedicated paragraph to models handling conflicting knowledge in context vs. in weights and how people have tested this/localized it in addition to the existing paragraph on model interventions for control (see "missing citations" section for some cites).
- Method: the limitation of only patching the outputs of the multi-head self-attention function could partially explain why, e.g., the authors fail to find any localization of the contextual answer mechanism in Figure 2d. I was surprised the authors chose to restrict their study to MHSA functions when there is substantial evidence that MLPs play some role in the production of parametric knowledge like factual queries. Given that we also know that there are redundant mechanisms in overparameterized LMs such that multiple causal interventions could lead to a large effect, I believe a more broad and impactful approach would be to first perform interventions at the final hidden state output of each layer (i.e., residual stream) before narrowing down to the MHSA or MLP functions.

More minor:
- heavy use of terms like "one dimensional subspace" to describe the impact of the paper's findings without any concrete definition given. The authors assume a substantial amount of linear algebra background in readers (see question below-- this may be due to my personal confusion, so I am open to resolving this critique in the rebuttal)
- method: the authors only focus on the final token position in their analysis, so they may be missing some interesting effects (such as which token positions the found causally relevant attention mechanisms at the final token position are attending to). However, I understand looking at more token positions can be quite computationally intensive and the experiments needed to be restricted in some way.
- It would be valuable or interesting to show some of the patchscope results in the paper apart from the best found set of layers in Figure 2, i.e., how the iterative algorithm built up to the final result.

**Questions:**

Questions:
- I don't fully understand the claim about the editing occurring in a low-dimensional/rank-one and orthogonal (to what?) subspace, when the learned square linear projection matrix ($P$) is of the same dimensions as the model hidden state. Is this referring to the fact that the intervention (eqn 4) only occurs on a vector defined by a singular constant? Even so, I still don't understand why the loss and/or editing rule (eqns 3 or 4) ensure that the learned projection matrix $P$ is rank-one or orthogonal. I would appreciate clarification on this in the rebuttal. (Regardless of my personal confusion, I think the paper could strongly benefit from at least some explanation of these terms for the reader's benefit).
- "We hypothesize that for a model to solve this task,..." (lines 69-70, 178-179): is this your hypothesis, or did it come from the prior work that you cite (Jin et al 2024)?
- Can you explain a bit more in the paragraph "Iteratively searching for model components" when you apply patchscopes vs. when you apply patching? I didn't understand in the pseudocode why you need to apply both when you could just directly calculate the score assigned to the token(s) of interest after activation patching as is traditionally done

Minor comments/suggestions:
- activation patching was first proposed in https://proceedings.neurips.cc/paper/2020/hash/92650b2e92217715fe312e6fa7b90d82-Abstract.html and https://arxiv.org/abs/2004.14623, not Meng et al 2022.
- it would be good to provide some small amount of background on what the "residual stream" is (and general structure of the Transformer block) for unfamiliar readers
- duplicated sentence in lines 060-065, typo line 130
- I didn't find the upper figures for each of 2a-2d to be particularly meaningful-- particularly if you're not patching *into* the residual stream directly, I didn't understand the dotted line
- a nit, but you use $a_c$ and $a_p$ for both predicted labels and correct labels. Later on you clarify that you only look at correctly predicted instances, so they are the same, but it would be worth mentioning this when you introduce the notation

Some missing citations:
- https://arxiv.org/abs/2402.11655
- https://arxiv.org/abs/2305.16572
- https://openreview.net/forum?id=auKAUJZMO6
- http://arxiv.org/abs/2404.06283

Some others about how models handle irrelevant context:
- http://arxiv.org/abs/2302.00093
- https://openreview.net/forum?id=Tigr1kMDZy
- http://arxiv.org/abs/2310.01558

---

> ### Author Response · Authors · 2024-11-20
> **Response Part 1**
>
> Thanks for your thorough engagement with our paper! We appreciate
> that you highlighted that our work is “well-motivated and of substantial interest”, “well-written/easy to follow”, with methods that are “particularly creative and interesting” and “generally comprehensive” experiments. We also appreciate your thoughtful feedback and questions; in this rebuttal, we hope to bring us toward the same understanding on the stated weaknesses.
>
> ## Weakness 1 “it overstates the novelty of proposing and providing evidence for the research question that there is a simple fundamental mechanism in LMs that modulates whether LMs defer to information in context vs in weights.”:
>
> We thank the reviewer for raising this important point and helping us better situate our work in the existing literature. We completely agree that it’s important to acknowledge their contributions towards understanding the mechanism for balancing context and prior knowledge. We have rephrased the relevant sections in the introduction to refer to the mentioned works and better reflect our contribution.
>
> We further want to acknowledge that we as a field are still far from fully understanding how exactly the underlying mechanism works. For example, it is not self-evident that “the same” mechanism always forms (even in different model architectures, trained on different datasets). We see our work as an important addition to the existing body of literature and understanding of the underlying mechanism by emphasizing that (1.) the mechanisms underlying the decision between context and prior knowledge in our CCS setting can be elegantly controlled in both directions with our single one-dimensional subspace, (2.) that we find similar mechanisms as the related work that mostly studied Pythia, GPT2, GPT-J, and Llama 2 also in a range of modern LLMs like Llama 3, Mistral and Gemma, (3.) that our one dimensional subspace transfers between base and fine tuned models.
>
> In summary, we thank you for your valuable input, which has led us to refine the wording of our contribution and our position in relation to related work. We believe this has improved the quality of our work and your feedback is greatly appreciated. Please let us know if the rewritten contributions and their novelty fits your understanding of the field now. Further, we propose rephrasing our focus as a search for a “fundamental mechanism in the form of a subspace”, to clearly differentiate our work from prior literature, but are also open to incorporating any alternative phrasing the reviewer might suggest here.
> ## Weakness 2 “Missing Related work”:
> Thank you for suggesting many related works. We have now (a) referenced your suggested works in the introduction and section 2, and (b) expanded on Yu et al 2023 and Ortu et al 2024 (and their shortcomings) to contextualize where our work fits in.
> ## Weakness 3 “Missing Investigation of MLPs and direct residual stream patching”:
> Thank you for your detailed feedback. We appreciate your suggestion to broaden our approach beyond just focusing on MHSA outputs, especially given the significant role that MLPs are thought to play in encoding factual knowledge.
> To clarify, our decision to concentrate on MHSA was guided by recent research (e.g. https://aclanthology.org/2023.emnlp-main.751.pdf, https://arxiv.org/pdf/2402.18154) which provides evidence that contextual answers are often processed at earlier token positions and later transported to the final token by attention heads. In fact, during our initial explorations, we did consider the role of MLPs but found that they didn't have a strong influence on the last token’s forward pass in our case. To clarify this, we’ve included more experiments on patching the MLP when investigating the prior answer in Appendix B that illustrate this limited impact.
> Regarding the lack of localization of the contextual answer mechanism: We believe our results do localize where the contextual answer is integrated into the residual stream, in particular in post layer 24 layers. The MHA prior to layer 24 do not seem to integrate information about the contextual answer. In Appendix Figure 7b, we highlight how layer 24 is pivotal, underscoring the emergence of the answer mechanism at this stage. This is especially interesting as we show that the intent is imported much earlier (12-16). The intent seems to be decoupled from how the context answer is integrated.
> Lastly, we acknowledge the appeal of interventions at the residual stream. While we tried this earlier on, it has some drawbacks. Intervening directly at the residual stream level overwrites existing information, which makes it difficult to isolate the contributions of specific components like MHSA or MLPs. By targeting these components individually, we gain finer insights into what each layer contributes. Nonetheless, we have put some additional experiments with patching the residual stream and showed that patching the MHA leads to a significantly better understanding.

---

> > ### Author Response · Authors · 2024-11-20
> > **Response Part 2**
> >
> > ## Minor Weakness 1 & Question 1: “Better explanation of the concept of a subspace”:
> > In the first paragraph of section 4.4, we added an informal intuition on the significance/meaning of a subspace encoding the context-vs-prior knowledge behavior. In Appendix H we also added an intuitive explanation about how a model’s residual stream representation can be decomposed into a sum of orthogonal components in different subspaces. This explanation accompanies equations 2a and 2b by visually illustrating (a) how the representation can be decomposed into a sum of the “context-vs-prior knowledge” subspace and the subspace containing “all other information” and (b) how you can patch in a different value in the “context-vs-prior knowledge” subspace to produce the patched residual stream representation.
> > Additionally, we have added a paragraph in Appendix G that explains in detail how the matrix P is parameterized and how we ensure its orthogonality. We hope this clears up confusion regarding editing in a rank-1 subspace and how we ensure the projection matrix is rank-1.
> > ## Minor Weakness 2 “Last token focus is limiting”:
> > We agree with the reviewer that this is a limitation. We intentionally made this choice to focus our investigation as otherwise there are exponentially more sites to look at. Further, our goal was to investigate the nature of how the model resolves the knowledge conflict - specifically the decision point. Building on other recent work and initial experiments, we deemed it plausible that this decision happens at the last token position - something which empirically validated to be a valid hypothesis. If the reviewer thinks that the subtitle “5.3.2 Where are a(q,eps) and a(q,c) integrated?” instead of “5.3.2 Where are a(q,eps) and a(q,c) computed?” would better fit this research focus, we will happily make that change.
> > ## Minor Weakness 3 and Question 3 “"Better explanation of the ‘iteratively searching for model components’ section including a visualization”:
> > This is an important point, and we recognize that our initial explanation may not have been sufficiently clear. With PatchScope, we can examine the model's internal state directly, rather than merely observing the patched output distribution. This represents a significant advancement over simply analyzing the probabilities of the tokens of interest at the model’s output. The PatchScope applied at the final layer (Layer 31) aligns with the patched output from a standard model forward pass.
> > The key problem that PatchScope addresses is the inability to detect internal changes when observing only the output state. If the model engages in self-healing — a process where later layers, which we refer to as late-suppression layers, mitigate the impact of the patching — the output distribution may not reveal the underlying effect. However, PatchScope enables us to identify these internal changes by exposing the impact of patching throughout the model, even when late-suppression layers obscure this effect at the output level.
> > Without PatchScope, we would be unable to pinpoint the base range effectively. If late-suppression layers are active, traditional methods based on logit differences would fail to detect the impact of patching. To assist reviewers in understanding this, we have included Figure 6 in the appendix. This figure provides a visual depiction of PatchScope at various stages of the algorithm, along with a detailed walkthrough of the algorithm. We hope this addition clarifies any confusion.
> > ## Question 1 "We hypothesize that for a model to solve this task,... (lines 69-70, 178-179): is this your hypothesis, or did it come from the prior work that you cite (Jin et al 2024)?”:
> > This is our hypothesis but we were clearly influenced by [Jin et al., 2024]. They have shown that there are separate sets of attention heads that dead with integrating contextual and prior information. Their paper does not talk about the step “deciding to answer with the context answer or the prior answer.“. Nonetheless, we have added a clearer attribution there. Thank you for pointing this out.
> >
> > # Minor comments
> > ## Activation Patching Citation
> > Thank you for raising this issue, we have implemented this.
> > ## Introduction of the Residual Stream Framework
> > We have added a chapter in Appendix D that properly defines the residual stream, and a reference to Appendix D in the main text’s first mention of the residual stream.
> > ## Usefulness of “Patching flow” in Figure 2
> > We agree here and have decided to remove this part from the figure, as the information can be displayed more efficiently. Thank you for pointing this out.
> > ## Definition of answer terms a_c and a_p
> > We have revised the formalization of the answers and they are now defined as a function of the query and either a context or the empty string. We think this is much cleaner. In general, this answer function always refers to the ground truth and never to the predicted answer. We hope that the new formalization is clear.

---

> ### Author Response · Authors · 2024-11-26
> **Follow-Up on Reviewer Feedback**
>
> We wanted to ask reviewer 1YMS if they have any questions about our response or updated submission. Please let us know if there is anything else that needs to be clarified or elaborated on to help with your concerns.

---

> > ### Comment · Reviewer_1YMS · 2024-11-26
> >
> > Thanks for the ping. I'm looking over your response and updated feedback now.

---

> > > ### Comment · Reviewer_1YMS · 2024-11-27
> > >
> > > Thank you for your detailed response and comprehensive changes to the paper. I particularly appreciate the addition of Appendices B-D and the additions to the revamped related works section. I think the rephrasing changes you've made about the novel nature of the mechanism you discover (subspace) seem valid and satisfy my initial concerns.
> > >
> > > My one remaining comment regarding contextualization w.r.t prior work is that you do discuss your dataset as a main contribution in the abstract + introduction, but you don't mention that very similar datasets (such as with conflicting contextual country-capital info) have been proposed and used in prior work, so it's unclear to me why you needed to create a new one. I think it would be good to at least acknowledge the dataset similarity in the related works when you discuss those papers (such as Yu et al 2023). Though I understand space is quite limited...
> > >
> > > As I mentioned in my initial review, I think the last token focus limitation is a reasonable scoping decision given the otherwise comprehensive nature of the experiments-- plus, I appreciated the detailed contextualization/explanation of this choice you now give in Appendix B.
> > >
> > > I have raised my score.

---

### Author Response · Authors · 2024-11-20
**General Response**

We sincerely thank all the reviewers for their time, feedback, and contributions to improving our paper. Below, we outline the general changes made to address your concerns, followed by individual responses to each reviewer.
To accommodate the page limit while incorporating new content, we have made the writing more concise. Further, we merged the original Sections 2 and 3, as both discussed related works. Consequently, the previous Section 4 is now Section 3, and so forth. All references in this response use the updated section numbers. Additionally, we refined our formalization and revised the notation of the answers to eliminate previous ambiguities and mathematical bugs, resulting in minor changes to the mathematical formulations in Section 3. We also added more experimental details about Gemma and Mistral in Appendices F and I.

In response to your reviews, we have implemented the following changes, all clearly marked in red text:
- A clearer positioning and contribution statement in the introduction.
- Integration of additional related work.
- An improved description of the algorithm in the appendix, including plots at intermediate states (Appendix A).
- A discussion about the MLPs and why our primary focus lies on the multihead attention (MHA) (Appendix B).
- Additional experiments for patching the residual stream directly (Appendix C).
- An explanation of the residual stream framework (Appendix D).
- A detailed explanation of how the orthogonal projection matrix P is parameterized (Appendix G).
- A primer on the mathematical intuition behind the orthogonal decomposition including a visualization (Appendix H).
- We have removed the patching flow from all the patching plots.
- A brief discussion of future work.

Other cosmetic changes which are not directly related to concerns raised by a reviewer (e.g., changing the motivating example in the introduction) are not highlighted in red.

We appreciate your constructive feedback and are confident that these revisions have strengthened our paper. Please refer to the individual responses below for more details.

---

### Meta-Review · Area_Chair_Wer4 · 2024-12-21

**Metareview:**

This paper identifies a linear subspace in a single layer that controls how strongly a model is sensitive to its context vs. prior knowledge.  By modifying activations at this layer, the paper achieves a "knob" in a model as well as base models from the same family. The model uses patching to answer three subquestions about where contextual and parametric answers are computed and what the "intent" is (prior knowledge vs. contextual knowledge).  Finally, the paper learns a projection matrix to project the intent into a vector that "steers" the model's activations towards a particular behavior mode.  This intervention works well on both instruction-tuned and base models, including base models where intent instruction does not work.

This paper explores an interesting idea in interpretability and brings a number of techniques to bear to localize context sensitivity within the model.  The results are stronger than one might expect, particularly the transfer between LLMs. Activation steering can be brittle, but these vectors actually seem to induce the right behavior somewhat broadly.

The main weaknesses of this paper are its practicality and its situation with respect to prior work. Both myC8 and Cdh2 wonder why such a knob would be used instead of fine-tuning. I think one has to buy the agenda of interpretability and assume that there will be benefits down the road to having lightweight intervention approaches like this, since this paper does not fully deliver them.

The second weakness is the relation to prior work. 1YMS mentions two prior papers which propose similar interventions for handling queries over counterfactual contexts.

While the weaknesses here are real, I believe the first weakness is shared by much of the model steering literature. In my view, this paper presents an interesting approach and gets it to work surprisingly well; I can imagine this methodology and results will be useful to other researchers in this area.

**Additional Comments On Reviewer Discussion:**

1YMS mentions two prior papers which propose similar interventions for handling queries over counterfactual contexts. The authors mostly push back on saying that their 1D subspace is new (and their work extends things to modern moddels, which in my view is not quite enough novelty). I believe this point should be better addressed in a revision.

myC8 and Cdh2 bring up the concern about why this method would be used. The authors describe how this could be more efficient and how interpretability can provide benefits.  I am moderately convinced by these; I think the ICLR community is interested enough in steering that even if the area has not delivered great results yet, it is a relevant and timely research question.

---

### Decision · Program_Chairs · 2025-01-22

Accept (Poster)